# Structural atlas of a human gut crassvirus

Oliver W. Bayfield[1✉], Andrey N. Shkoporov[2], Natalya Yutin[3], Ekaterina V. Khokhlova[2], Jake L. R. Smith[1], Dorothy E. D. P. Hawkins[1], Eugene V. Koonin[3], Colin Hill[2] & Alfred A. Antson[1✉]

CrAssphage and related viruses of the order *Crassvirales* (hereafter referred to as crassviruses) were originally discovered by cross-assembly of metagenomic sequences. They are the most abundant viruses in the human gut, are found in the majority of individual gut viromes, and account for up to 95% of the viral sequences in some individuals[1–4]. Crassviruses are likely to have major roles in shaping the composition and functionality of the human microbiome, but the structures and roles of most of the virally encoded proteins are unknown, with only generic predictions resulting from bioinformatic analyses[4,5]. Here we present a cryo-electron microscopy reconstruction of *Bacteroides intestinalis* virus ΦcrAss001[6], providing the structural basis for the functional assignment of most of its virion proteins. The muzzle protein forms an assembly about 1 MDa in size at the end of the tail and exhibits a previously unknown fold that we designate the 'crass fold', that is likely to serve as a gatekeeper that controls the ejection of cargos. In addition to packing the approximately 103 kb of virus DNA, the ΦcrAss001 virion has extensive storage space for virally encoded cargo proteins in the capsid and, unusually, within the tail. One of the cargo proteins is present in both the capsid and the tail, suggesting a general mechanism for protein ejection, which involves partial unfolding of proteins during their extrusion through the tail. These findings provide a structural basis for understanding the mechanisms of assembly and infection of these highly abundant crassviruses.

Viruses infecting bacteria are thought to be the most abundant biological entities on earth. Until recently, all such viruses were identified after growing on cultivatable bacterial hosts, but these are likely to account for less than 5% of the actual global diversity[7–9]. Recent analyses of viral 'dark matter', made possible by cross-assembly of metagenomic data, have led to the discovery of many new viruses, including crAssphage—the most abundant virus in the human microbiome—and related viruses comprising the newly established order *Crassvirales*[2,4,5,10]. These viruses have been closely associated with human populations throughout evolution[11]. Crassviruses appear to infect exclusively diverse members of the bacterial phylum *Bacteroidota*[3,8,11,12]. In healthy adult Western cohorts, crassviruses are detected in 98–100% of individuals, often dominating the faecal virome[3]. The virus families Flandersviridae, Quimbyviridae and Gubaphage have been identified as close contenders with the *Crassvirales* in terms of their detection frequency[8,12]. However, *Crassvirales* appear to be unique in their showing both a high abundance in individual viromes and across varied cohorts globally[3,8,11,12]. Some crassviruses appear to establish a form of carrier-state infection, with delayed lysis of the infected bacteria, and appear to follow a piggyback-the-winner virus–host dynamic[6,13]. How this lifestyle contributes to their prevalence and abundance remains unclear.

Virus ΦcrAss001 (*Kehishuvirus primarius*) was the first representative of this order to be isolated in pure culture from human faecal samples[6]. Here, we present the characterization of ΦcrAss001 using cryo-electron microscopy (cryo-EM), enabling de novo structure building and

elucidation of functions for around 1,440 protein subunits constituting the virion, leading to the identification of a novel 'crass fold' in the distal part of the tail. The structure reveals a mechanism by which a large complement of virally encoded cargo proteins, representing a total molecular mass of approximately 5.5 MDa, can be stored in the capsid and tail. The presence of one such cargo protein inside both the capsid and tail suggests a mechanism for protein ejection that takes place before complete ejection of genomic DNA and involves partial unfolding of cargo proteins. Sequence analysis revealed the extent of conservation of virion proteins throughout the *Crassvirales*, providing a structure–function atlas for this viral order.

## Linking virion and genome architectures

The structure of the ΦcrAss001 virion was determined using single-particle 3D reconstructions from electron microscopy data (Fig. 1a,b). Reconstructions were determined to a resolution that enabled building atomic models for most of the proteins that constitute the virion (Fig. 1c). The virion possesses a short non-contractile tail extending approximately 44 nm from an icosahedral capsid. Inside the capsid, double-stranded DNA (dsDNA) is packed with a left-handed supercoiling (Fig. 1c, left), and at high density leading to hexagonal close-packing (Fig. 1c, right). Moreover, the reconstruction reveals the structural hallmark proteins common to all crassviruses, and sequence analysis enabled mapping of the virion proteins that are most broadly conserved across this viral order (Fig. 1d). In particular, no homologues

[1]York Structural Biology Laboratory, Department of Chemistry, University of York, York, UK. [2]APC Microbiome Ireland and School of Microbiology, University College Cork, Cork, Ireland. [3]National Center for Biotechnology Information, National Library of Medicine, National Institutes of Health, Bethesda, MD, USA. ✉e-mail: oliver.bayfield@york.ac.uk; fred.antson@york.ac.uk

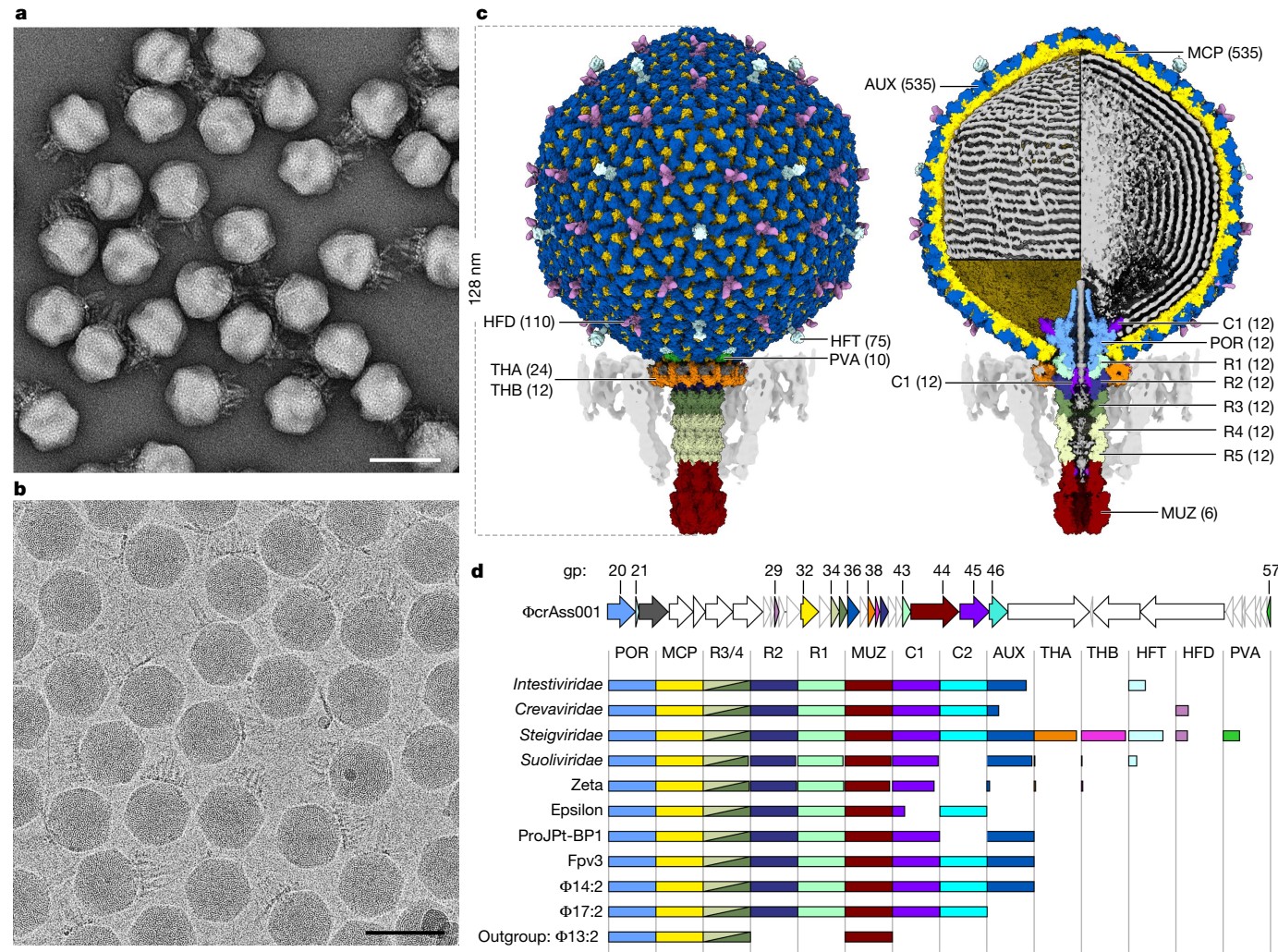

**Fig. 1 | Structure of the ΦcrAss001 virion and functional assignments.**
**a**,**b**, Electron micrographs of virions recorded using negative staining (**a**; n = 16 micrographs) and under cryogenic conditions (**b**; n = 44,006 micrographs). Scale bars, 100 nm. **c**, Molecular surface of virion reconstruction viewed from outside (left) and z-clipped (right), coloured by gene product. DNA (grey) on the z-clipped image is depicted either as an outer layer lining the capsid wall (top left) or clipped (right), or is removed (bottom left). Unmodelled regions are displayed as map, lowpass filtered to 8 Å (parts of head fibre proteins and DNA) and to 16 Å (tail fibres, semi-transparent grey). AUX, auxiliary capsid protein; C1 and C2, cargo proteins; MUZ, muzzle protein; POR, portal protein; PVA, portal vertex auxiliary capsid protein; R1–R4, ring proteins; THA and THB, tail hub proteins. Numbers in brackets indicate the protein copy number in the virion. **d**, Top, the ΦcrAss001 genome (accession: NC_049977.1) region 11643–68717 showing open reading frames as arrows coloured corresponding to gene products (gp) in **c**, scaled according to gene length. Bottom, gene conservation, with the length of each bar representing the fraction of each crassvirus group in the order *Crassvirales* containing a detectable homologue, coloured corresponding to **c**. Names on the left refer to the virus families (*Intestiviridae*, *Crevaviridae*, *Steigviridae* and *Suoliviridae*) and groups (zeta and epsilon) with individual members from non-human gut environments represented by ProJPt-BP1 (termite gut), Fpv3 (fish farm) and Φ14:2, Φ17:2 and Φ13:2 (marine environments).

of the capsid auxiliary protein and the tail muzzle protein were detected among the currently characterized structures of viral proteins.

## The capsid proteins

The icosahedral reconstruction of the capsid, which was determined to a resolution of 3.01 Å (Extended Data Table 1), has an outer diameter of approximately 88 nm. The capsid has a triangulation number ($T$) of nine, with an asymmetric unit comprising nine copies of the major capsid protein (MCP) gp32, nine copies of the capsid auxiliary protein gp36, one copy of the trimeric head fibre protein (HFT) gp21, and two copies of the dimeric head fibre protein (HFD) gp29 (Fig. 2a).

The MCP adopts the HK97-like fold—the canonical fold of the capsid proteins of the viruses in the class *Caudoviricetes*[14], with the main body comprising the A-domain and P-domain (Fig. 2b). A β-barrel insertion in the E-loop, composed of residues 79–180 (I-domain), is a

widespread feature of the crassviruses[5]. This insertion domain forms a three-fold symmetric assembly with the auxiliary protein at the capsid surface (Fig. 2a and Extended Data Fig. 1), making up much of the exposed surface of the capsid and contributing substantially to the capsid wall thickness and stability. An insertion at a similar position of the E-loop, albeit of a different fold (comprising one helix and six strands), is present in the major capsid protein of bacteriophage T4[15]. The ΦcrAss001 I-domain is structurally similar to the I-domain of the major capsid protein of phage P22 (DALI Z-score of 6.1 and root mean square deviation (r.m.s.d.) of 2.7 Å with Protein Data Bank (PDB) entry 5UU5 chain E). However, in P22, this domain (residues 226–344) is part of the A-domain, rather than the E-loop.

The auxiliary protein gp36 has a two-domain architecture (Fig. 2c). The PH domain is composed of residues 1–74 and 238–333 and shows structural similarity to the pleckstrin homology-like domain (DALI Z-score of 5.7 and r.m.s.d. of 3.2 Å with PDB entry 4DX9, belonging to

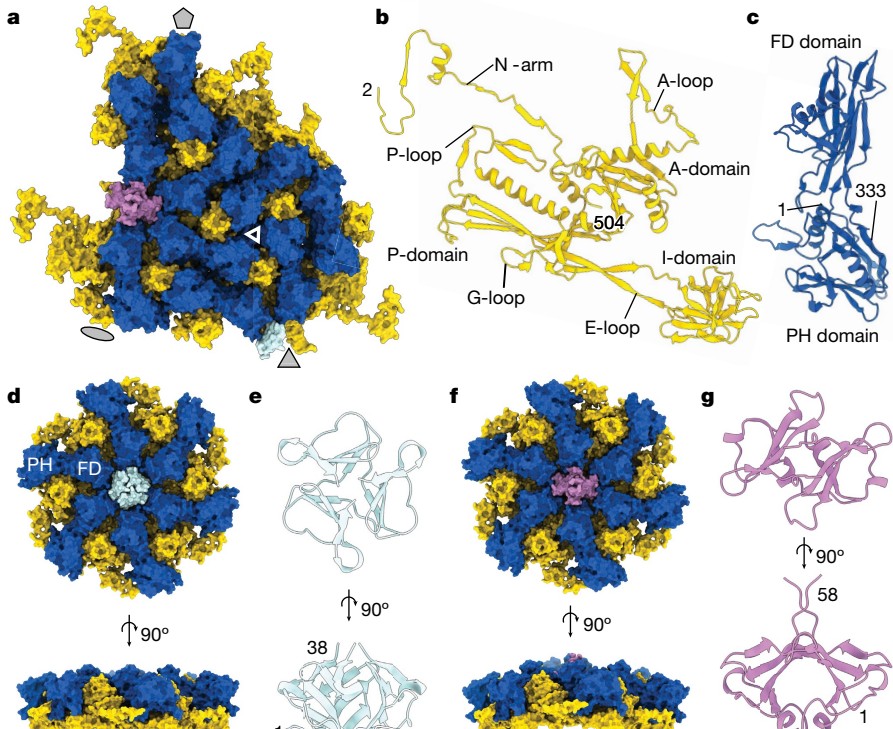

**Fig. 2 | ΦcrAss001 capsid proteins. a**, Molecular surface of the asymmetric subunit of the icosahedral capsid with indicated symmetry axes (grey shapes), with major capsid protein gp32 (yellow), capsid auxiliary protein gp36 (dark blue), head fibre proteins gp21 (HFT; pale blue, trimer) and gp29 (HFD; pink, dimer). The triangle indicates a local quasi-$C_3$ symmetry formed by I and PH domains. **b,c**, Ribbon diagrams of the major capsid protein subunit (**b**) and the capsid auxiliary protein gp32 (**c**). **d**, Molecular surface of the $C_3$ capsid hexon. **e**, Ribbon diagram of the HFT trimer. **f**, Molecular surface of the skewed hexon. **g**, Ribbon diagram of the HFD dimer. Numbers indicate terminal residues of the model.

CATH superfamily 2.30.29.30). The FD domain is composed of residues 75–237 and is structurally similar to the *Helicobacter pylori* flagellar cap protein FliD (DALI *Z*-score 7.3 and r.m.s.d. 3.4 Å with PDB entry 6IWY chain A) and *Pseudomonas aeruginosa* flagellin FliC (DALI *Z*-score 6.7 and r.m.s.d. 3.2 Å with PDB entry 4NX9 chain A). To our knowledge, auxiliary proteins with such a fold and size have not been observed in viral capsids, and they represent a unique feature of most of the crassviruses (Fig. 1d).

The FD domain of the auxiliary protein interacts with the A-domain of the MCP, around the periphery of each hexamer and pentamer. Each hexamer can adopt one of two conformations: winged or planar (Fig. 2d,e). Winged hexons span the edges of the icosahedron, on both sides of the icosahedral two-fold symmetry axes, whereas planar hexons are positioned at the three-fold symmetry axes. At the centre of each hexon is a pocket formed by the MCP and auxiliary protein, which serves as a binding site for fibre proteins. A trimer of the head fibre protein gp21 (residues 1–38) binds in the planar hexon pocket at the three-fold symmetric site (Fig. 2d), whereas a dimer of the capsid fibre protein gp29 (residues 1–58) binds in the winged hexon (Fig. 2e). Thus, the *T* number of the capsid is responsible for generating two distinct hexon-pocket environments, which are specific to two different oligomeric fibres. These two fibres are likely to afford two different binding specificities to external molecules located either on the surface of the bacterial host cell or on the surface of the gut. Homologues of these fibres are present in *Intestiviridae* (alpha), *Steigviridae* (beta) and *Suoliviridae* (delta) groups of crassviruses in the case of the HFT gp21, and in the *Steigviridae* (beta) and *Crevaviridae* (gamma) groups in the case of the HFD gp29 (Fig. 1d). Homologues of the C-terminal domain of gp21 (residues 42–97) with high sequence similarity were detected in viruses outside of the *Crassvirales*, including gp8.5 of *Bacillus* phage ϕ29 (40% identity with residues 226–280, and DALI Z-score of 4.3 and

r.m.s.d. of 3.2 Å between PDB 3QC7 chain A and gp21 AlphaFold structure prediction) (Extended Data Fig. 2). Gp8.5 has been suggested to bind to cell wall factors of Gram-positive bacteria[16]. AlphaFold modelling showed that the C-terminal domain of gp29 (residues 58–118) forms a β-sandwich (Extended Data Fig. 2) with structural similarity to the immunoglobulin-like fold (DALI *Z*-score of 6.8 and r.m.s.d. of 2.1 Å with PDB 2NSM chain A; CATH superfamily 2.60.40.1120). A DALI search for the N-terminal domains of gp21 and gp29 did not detect any significantly similar structures. Although most groups of crassviruses did not appear to contain homologues of gp21 and gp29, as yet uncharacterized proteins might perform analogous roles.

## The portal vertex

One vertex of the capsid contains the dodecameric oligomer of portal protein gp20. The portal protein is considered the nucleation point for capsid assembly and serves as the docking site for the tail[17–20]. The portal is surrounded by the five-fold symmetric capsid vertex, contacting 10 major capsid protein subunits. The majority of the portal protein oligomer exhibits 12-fold symmetry and the localized reconstruction imposing $C_{12}$ symmetry enabled structure determination at 3.09 Å resolution (Fig. 3). The majority of the portal protein residues, except for some flexible loop segments (33–35, 68–71, 269–324 and 550–563), could be confidently modelled (Fig. 3a,b). The portal protein exhibits a canonical fold[21] (Fig. 3a), but with a long C-terminal barrel of the oligomer resembling the corresponding domain of the P22 portal protein[22].

At the top of the portal protein, an extended α-helical barrel (C-terminal residues 672–739) extends around 80 Å towards the core of the capsid (Fig. 3a). This extended feature has an internal van der Waals diameter of around 45 Å at the base of the crown and around 25 Å

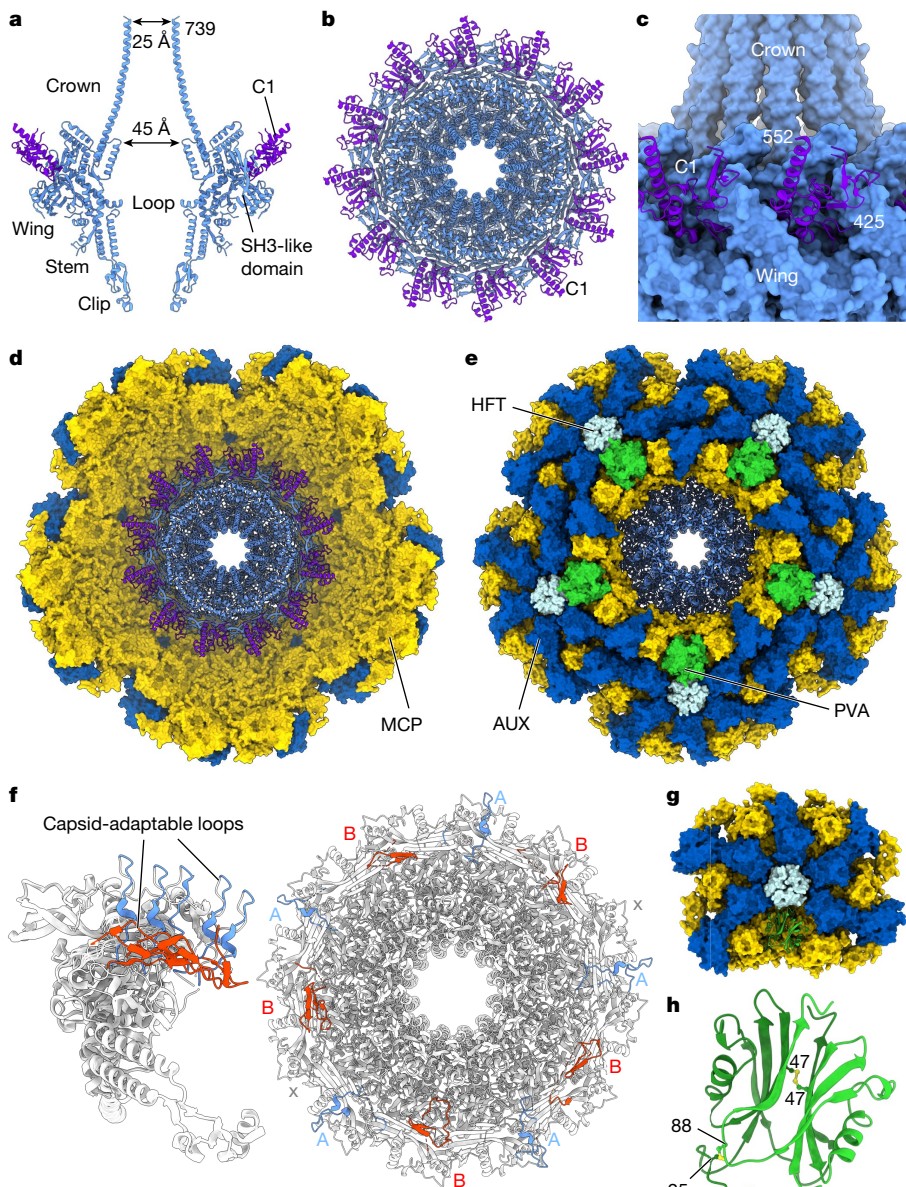

**Fig. 3 | The portal protein and the surrounding capsid vertex. a**, Two opposing subunits of the portal protein (light blue) and the C1 cargo protein (purple) depicted as ribbons. **b**, Dodecameric assembly of the proteins shown in **a**, viewed along the channel axis. **c**, The portal protein wing region (light blue) interacting with C1 cargo protein subunits (purple). **d**,**e**, Molecular surface of the portal-containing capsid vertex, comprising major capsid protein gp32 (yellow), auxiliary capsid protein gp36 (dark blue) and portal protein gp20 (ribbon, light blue), viewed from inside the capsid along the central channel (**d**) and from outside the capsid (**e**) with the portal vertex auxiliary protein gp57 in green and the head fibre protein gp21 in pale blue. **f**, Ribbon diagram of the portal protein, depicting 12 subunits superimposed (left) and the oligomer viewed along the central axis (right) with capsid-adaptable loops shown in light blue and red corresponding to conformations A and B, respectively. **g**, Unit of the $C_5$ reconstructed portal-containing capsid vertex, with the gp57 (PVA) protein dimer shown as green ribbon. **h**, The gp57 dimer with inter-subunit disulfide bonds show in yellow. Numbers indicate terminal residues of the model.

at its exposed end. This barrel is well positioned to direct DNA into the capsid during packaging, so that DNA enters at the central part of the capsid towards the opposite capsid wall.

The main body of the portal protein beneath the Crown domain is made up of the stem domain, composed of α-helices defining its central tunnel, and the wing domain (Fig. 3a). Tunnel loops, which are not defined in maps owing to their apparent flexibility, connect the inner stem helix (residues 519–541) with a 34-residue-long kinked helix that is part of the wing (Fig. 2b). These tunnel loops (residues 550–563) are likely to form a narrow constriction in the portal tunnel, as observed in portal proteins of other viruses[18,21,23–25].

At the surface of the portal wing, there is clear density corresponding to residues 425–552 of the gp45 cargo protein C1 (Fig. 3b,c). Twelve such segments of gp45 interdigitate with wing loops (SH3-like domains) of the portal, residues 337–380 (Fig. 3b,c).

The 12-fold symmetrical portal oligomer sits within the 5-fold symmetrical capsid vertex (Fig. 3d,e). The localized asymmetric reconstruction of the portal–capsid vertex (obtained without imposing symmetry), determined at 3.60 Å resolution, showed that a loop at the peripheral area of the wing domain, residues 268–319, is flexible and can adopt two preferred conformations (types A and B) in interacting with the $C_5$-symmetric capsid vertex (Fig. 3f). In the type A conformation,

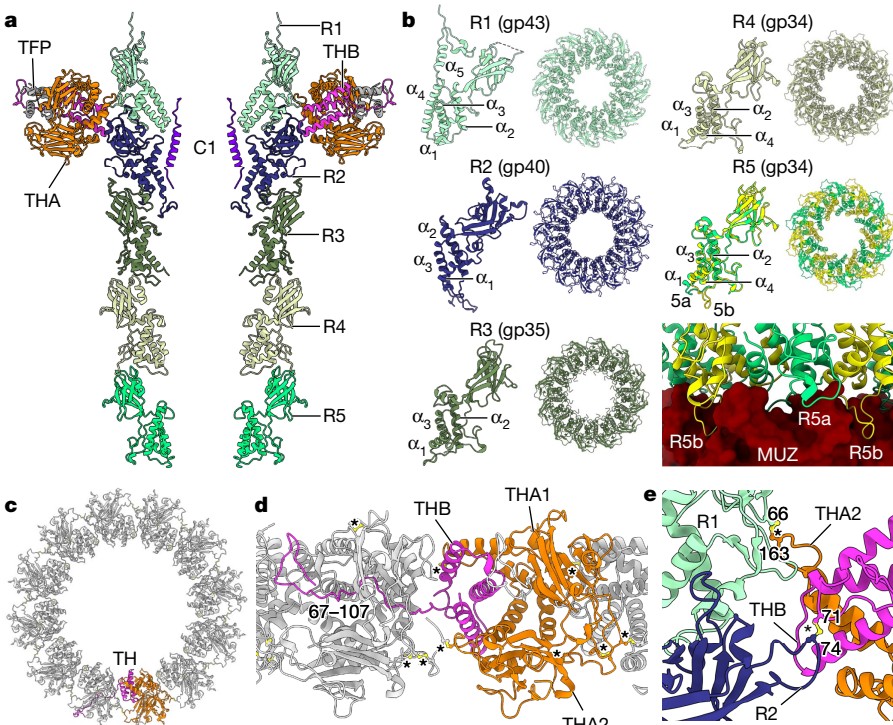

**Fig. 4 | The tail barrel and fibre docking hubs. a**, Two opposing chains of ring proteins of the tail barrel, depicted as ribbon diagram, from top to bottom: R1 (gp43) (light cyan), R2 (gp40) (navy), R3 (gp35) (mid green), R4 (gp34) and R5 (gp34) (light green), THA (gp38) (orange), THB (gp39) (pink) and gp22 (dark grey). **b**, Individual subunits of each ring protein alongside their dodecameric ring assemblies viewed along the central axis. **c**, The collar formed by 12 tail hub assemblies, viewed along the central axis from the capsid end, depicted in ribbon diagram with chains of one tail hub coloured orange for THA1 and THA2 and pink for THB. TH, tail hub. **d**, The collar as in **c**, rotated 90° and enlarged, showing interlocking tail hubs. **e**, Enlarged view of disulfide bonds (yellow, ball and stick) between tail hub proteins and R1 and R2 proteins. Asterisks designate disulfide bonds.

residues 277–290 of this 'capsid-adaptable loop' are oriented radially away from the portal's tunnel axis, whereas in the type B conformation residues 292–306 of the loop are arranged tangentially to the portal's circumference (Fig. 3f). In 2 of the 12 subunits of the portal protein, these loops were not resolved in either conformation (denoted 'x'), resulting in the A-B-x-A-B-A-B-A-x-B-A-B order of the conformational states, counting clockwise if viewed from outside of the capsid (Fig. 3f). The result is a pseudo-five-fold symmetrical arrangement of capsid-adaptable loops around the 12-subunit oligomer, assisting the formation of optimal interactions with the five-fold symmetric capsid vertex.

The surrounding capsid vertex was also reconstructed with $C_5$-symmetrical averaging to a resolution of 3.30 Å (Fig. 3g). The portal oligomer positioned in the vertex occludes the binding site of the capsid auxiliary protein gp36, which is substituted by two copies of the portal vertex specific auxiliary protein, gp57 (Fig. 3g,h). The gp57 dimer has approximate two-fold symmetry, reinforced by disulfide bonds between residues 47–47 and 25–88 of the two subunits (Fig. 3h). Notably, hexameric capsomers bordering the unique vertex contain trimer capsid fibres of gp21 (Fig. 3e), in contrast to the hexamers bordering other pentameric vertices, which contain dimeric capsid fibres of gp29.

## Tail barrel assembly and fibre hubs

The portal protein's clip domain interacts with the tail barrel, which extends approximately 23 nm towards the muzzle protein complex at the tip of the tail (Fig. 4). The barrel was reconstructed by imposing $C_{12}$ symmetry to a resolution of 3.20 Å, and with $C_6$ symmetry to a resolution of 3.30 Å. The barrel is composed of five stacked dodecameric 'rings', termed ring 1 (R1, at the portal end) to ring 5 (R5, at the muzzle end) (Fig. 4a). These structures are formed by four paralogous ring proteins: R1, R2 and R3 are encoded by genes gp43, gp40 and gp35, respectively, whereas R4 and R5 are both encoded by gp34 (Fig. 4b). Such a multi-ring barrel structure results in ΦcrAss001 having a much longer tail than typically observed in other podoviruses, which have only been observed to possess one such ring formed of one protein. The ring proteins share highly similar folds and consist of two domains, with three structurally conserved α-helices and a β-sandwich domain (Fig. 4b). In R5 (gp34), residues 15–26 adopt 2 conformations—5a and 5b—that facilitate binding to the 6-fold symmetric muzzle complex (Fig. 4b). Despite their conserved fold, the sequence identity among the ring proteins is relatively low, ranging from 14.9% (R2–R3) to 18.6% (R1–R2), although the more closely related paralogues in ΦcrAss001—R3 and R4—are 30.6% identical.

The α-helical domain of gp34 shares a similar fold with tail-joining factors in tailed bacteriophages with different tail morphologies (HK97, Mu and P22) and in gene-transfer agents[26,27]. The presence of multiple ring protein genes, however, appears to be a conserved and defining feature of crassviruses (Fig. 1d). Some variation exists in this pattern of conservation: genes 34 (R3) and gp35 (R4 and R5) of ΦcrAss001 are more recent paralogues, and many other crassviruses instead encompass a single homologue of R3, R4 or R5, and therefore together with R1 and R2 possess three ring protein genes. The extended tail barrel in ΦcrAss001 that results from this set of ring proteins may assist the tail muzzle in reaching the outer cell membrane, through the approximately-30 nm compact capsular polysaccharide[13]. Having three ring protein genes rather than four could result in a shorter tail barrel, and the outgroup in sequence analysis, phage Φ13:2, is distinctive in its possessing only the R3/R4 gene homologue (Fig. 1d).

R1 and R2 form an interface for interaction with the tail fibre docking hubs, which are the point of attachment of the tail fibre appendages

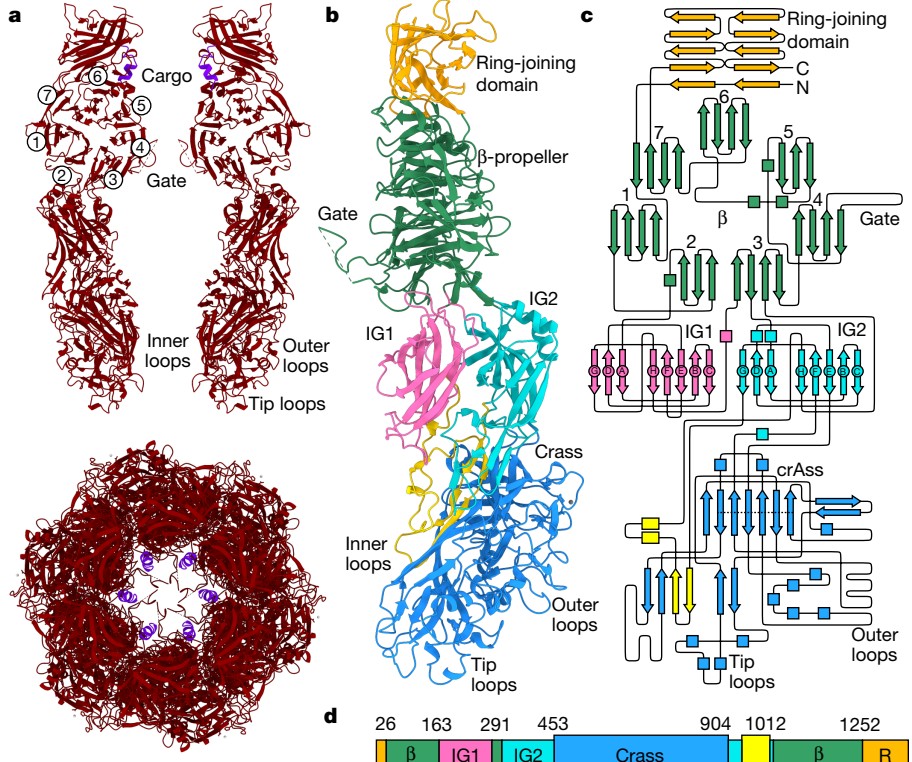

**Fig. 5 | The muzzle assembly. a**, Top, two opposing subunits of the muzzle protein shown as ribbons (red). Bottom, a hexamer of the muzzle viewed along the central axis with cargo protein in purple. Circled numbers denote the blades of the β-propeller. **b**, A single subunit of the muzzle protein (right chain in **a** rotated by 90° around the central tunnel axis), with the ring-joining domain in orange, the β-propeller domain in green, the immunoglobulin-like domain 1 (IG1) in pink, IG2 in cyan and the crass domain polypeptide segments in light blue and yellow. **c**, Topology diagram of the muzzle protein with domains coloured as in **b**. The dotted line in the crass domain indicates the β-barrel strands. **d**, Schematic of the muzzle protein with individual domains coloured as in **b**,**c**, with domain boundary residue numbers indicated. β, β-propeller domain. R, ring-joining domain.

(Fig. 4a,c). The tail hub (TH) proteins form a continuous 'collar' around the tail barrel (Fig. 4c). Each hub assembly is an $A_2B$ trimer that consists of two copies of THA (gp38) and one copy of the smaller protein THB (gp39) (Fig. 4d). The C-terminal segment of THB, residues 67–107, extends into the neighbouring hub, resulting in continuous interlocking ring of 12 docking hubs around the barrel (Fig. 4d). Each docking hub, in turn, interacts with the trimeric tail fibre protein gp22, accommodating its N-terminal residues 1–26 (Fig. 4a). Gp22 interacts with additional tail fibre proteins making up the inner and outer tail fibre assemblies, which are highly flexible and less well resolved in localized reconstructions. The tail barrel and fibre docking hubs are stabilized by disulfide bonds (Extended Data Table 2), notably between fibre docking hub proteins (Fig. 4d), the fibre docking hub and ring proteins (Fig. 4e), and between R1 and R2.

## The muzzle assembly

At the end of the tail is a large hexameric assembly of the muzzle protein gp44 (Fig. 5). The muzzle protein is composed of five domains (Fig. 5a,b), of which the ring-joining domain and the seven-bladed β-propeller are similar with those of the phage T7 nozzle protein[19], and the remaining three domains are distinct. Below the β-propeller are two immunoglobulin-like domains, and the fifth domain comprising the bulk of gp44 has no counterpart in known virus tail proteins or in any structurally characterized proteins.

Inserted between blades 2 and 3 of the propeller is immunoglobulin-like domain 1 (IG1, residues 163–292) and inserted into blade 3 is an immunoglobulin-like domain 2 (IG2); IG1 and IG2 share a similar topology (Fig. 5b). This immunoglobulin-like fold is structurally similar to the C2-set domains of immunoglobulins[28] (CATH superfamily 2.60.40.150), albeit with a different order of strands, with highest structural similarity observed for synaptotagmin-1[29] (DALI Z-score of 5.1 and r.m.s.d. of 3.9 Å with PDB 5CCH chain F).

Serving as a platform for assembly of the large domain forming the distal part of the muzzle is IG2 (between residues 318–1021). This domain is formed by insertions into loops between IG2 strands E–F (residues 453–904) and G–H (residues 940–1012), with the E–F insertion making up the bulk of this region (Fig. 5c). As this domain adopts a fold with no significant structural similarity to known protein domains, we refer to it here as the 'crass fold'. A small β-barrel below domain IG2 (indicated by a dotted line on Fig. 5c; 70 residues spanning segments 453–462, 563–571, 577–580, 581–591, 709–716, 725–733, 833–839, 894–901 and 902–905) structurally resembles other 6-stranded β-barrels (for example, r.m.s.d. of 3.2 Å with 50S ribosomal protein L2, PDB code 4U67, chain B).

The muzzle is gated by a loop within blade 4 of the propeller (Fig. 5a,b). One role of the muzzle is likely to be the formation of this constriction, preventing leakage of DNA and proteins contained in the capsid and tail until infection is initiated. Owing to the restricted channel diameter above and below the gate, and interactions at the inter-subunit interfaces, opening it would require conformational rearrangements of the IG and crass domains. The crass domain, owing to its location on the tail tip and the presence of multiple exposed surface loops, is likely to have a role in contacting the cell surface, thus initiating these conformational changes. The muzzle protein is highly conserved among the crassviruses (Fig. 1d), but exhibits length variations resulting from insertions into the crass domain and IG2 domain regions that can affect the assembly's surface properties.

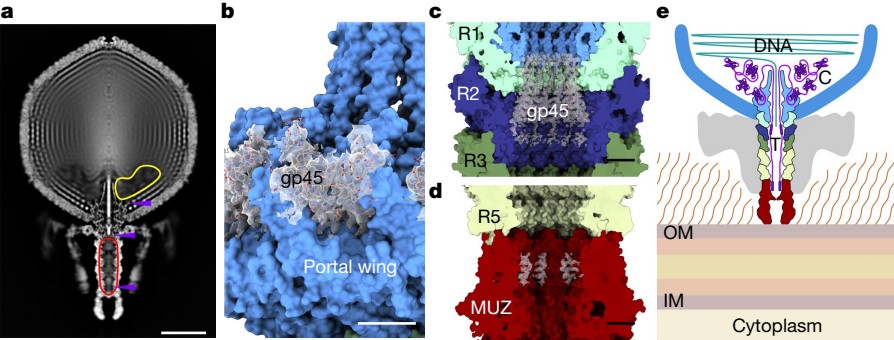

**Fig. 6 | The cargo protein zones. a**, *Z*-clipped view of the rotationally-averaged virion map. The protein cargo zones are indicated by outlines inside the capsid (yellow) and in the tail barrel (red). Purple arrows indicate locations of structured regions of cargo proteins that are resolved in reconstructions. **b–d**, Structures of the portal-bound domain of C1 (gp45) (**b**), the tail barrel-bound region of C1 (**c**) and the muzzle-bound cargo protein fragment (**d**) are shown with the corresponding density maps (transparent grey). **e**, Schematic of cargo protein locations inside the virion. Proteins (purple) are stored in the cargo zones located in the tail (T) and capsid (C). IM, inner membrane; OM, outer membrane.

## Cargo protein storage zones

Crassviruses are expected to carry several viral gene products inside the virions, including the large multi-subunit RNA polymerase[30], collectively referred to here as cargo proteins. Consistent with this expectation, mass spectrometry of ΦcrAss001 virions (Extended Data Table 3) detected the presence of proteins referred to here as cargo protein 1 (gp45, 90.8 kDa), cargo protein 2 (gp46, 60.2 kDa), and three RNA polymerase-associated proteins: gp47 (262.5 kDa), gp49 (154.5 kDa) and gp50 (276.9 kDa), which are broadly conserved among the crassviruses (Fig. 1d).

To investigate how these cargo proteins pack inside the virion, despite the absence of clear corresponding density in reconstructions, rotational averaging of the virion map around the axis of the portal protein tunnel was performed (Fig. 6a). Inspection of this map revealed an area inside the capsid with a substantially lower density where packing of the surrounding layers of DNA was discontinuous. This internal capsid region located close to the portal vertex was estimated to be about $1.10 \times 10^7$ Å$^3$ in volume. Also notable is the density inside the tail barrel, which is higher than would be expected from solvent alone, and a strong rod-like feature within the portal (Fig. 6a). The space inside the tail barrel measures approximately $8.15 \times 10^5$ Å$^3$.

In the cryo-EM reconstructions, fragments of cargo protein 1 (C1) (gp45), were resolved inside both the capsid and the tail. As noted, residues 425–552 fold into a compact domain bound to the portal protein inside the capsid (Fig. 6b). Unexpectedly, well-defined density at the inner surface of R1 and R2 of the tail barrel corresponds to 12 copies of an additional α-helical segment of the same protein, gp45 (residues 214–245) (Fig. 6c). Thus, different portions of gp45 are found inside either the capsid or the tail, separated by the portal protein tunnel.

On the basis of volume measurements, we can estimate the storage capacity of the cargo zones. Assuming proteins were packed at 2.15 Å$^3$ per Da volume per molecular weight, consistent with typical protein crystals with around 43% solvent content[31], the capsid and tail cargo zones could store proteins with a total molecular mass of up to around 5.1 MDa inside the capsid and an additional 0.4 MDa inside the tail barrel. These two zones can thus harbour multiple copies of the cargo proteins, as detected by mass spectrometry (for example, 6 copies of each the three RNA polymerase-associated protein and up to 12 copies of the smaller proteins gp45 and gp46). The remaining internal capsid volume available for the 102,679 bp genome is approximately $1.84 \times 10^8$ Å$^3$, resulting in a DNA packing density of 0.56 bp nm$^{-3}$, consistent with packaging densities observed for other viruses[32].

A further region of density inside the tail was observed at its distal end, next to the ring-joining domain of the muzzle protein (Fig. 6d). This region corresponds to 6 copies of a 14-residue segment containing an α-helix. Although this segment could not be assigned to a specific protein, it probably corresponds to yet another N-terminal segment of gp45, extending from the 214–245 fragment. Indeed, peptides spanning the regions 79–101 and 181–190 of gp45 were detected by mass spectrometry (Extended Data Fig. 3). Consistent with this, the $8.15 \times 10^5$ Å$^3$ volume of the tail cargo zone could accommodate up to about 0.4 MDa of protein material, which is sufficient for 12 copies of residues 1–245 of gp45 (about 306 kDa). This makes the capsid cargo zone the likely locus for the remaining cargo proteins, C2 (gp46) and the RNA polymerase-associated proteins (gp47, gp49 and gp50) (Fig. 6e).

## A model for protein ejection

The data reported here, along with previous observations on phage P22, and other podoviruses including T7, provide the basis for proposing a hypothetical model for protein ejection (Extended Data Fig. 4). Although podoviruses do not have a tail barrel as extended as that of ΦcrAss001, many do contain 'ejectosome' proteins that are packed as cargo inside their capsids. In the bacteriophage P22 capsid, there are 3 such proteins[33] (gp7, gp16 and gp20) with a combined mass of around 2.5 MDa. For T7, ejection proteins were shown to form a membrane-spanning channel during the initial phase of infection[34]. Although not all cargo proteins were resolved in the reconstructions of ΦcrAss001, these proteins have to exit the capsid ahead of DNA because they form the channel for DNA ejection. As in the case of T7, we expect that gp45 and gp46 of ΦcrAss001 are involved in transmembrane channel formation, with the other three cargo proteins annotated as RNA polymerase-related proteins[5]. Given that one region of gp45 (residues 214–245) is already poised for ejection, positioned inside the tail barrel of ΦcrAss001, this protein is likely to be ejected first and to contribute to channel formation. A transmembrane helix is predicted by TMHMM-2.0[35] to exist in the N-terminal region of gp45 (residues 136–158) (Extended Data Fig. 3), which could insert into the host cell membrane in a similar manner as described for the T7 protein gp14[36].

The two parts of cargo protein C1, gp45, identified in the tail barrel (residues 214–245) and in the capsid (residues 425–552) are separated by a distance of approximately 420 Å and could be bridged by the connecting 179 residues. Indeed, peptides spanning the regions 275–300 and 307–320 of gp45 were detected by mass spectrometry (Extended Data Fig. 3), indicating that these regions are present inside the virion. We can expect gp45 to be ejected in the N→C direction: with the N-terminal part located in the tail exiting first followed by the C-terminal domain located in the capsid (Extended Data Fig. 4). The size and globular structure of the C-terminal part of gp45 (Fig. 6b) suggests that its ejection involves partial unfolding.

The other four proteins detected by mass spectrometry (gp46, gp47, gp49 and gp50) are predicted by AlphaFold2[37] to adopt folded α-helical structures (Extended Data Fig. 5) and consistently, the crystal structure of a homologue of gp49 showed it is predominantly α-helical[30]. Several constrictions inside the tail barrel are less than 50 Å in diameter, requiring that these cargo proteins are partially unfolded to pass through.

Although understanding precisely how proteins and DNA are ejected requires further studies, it is likely that ejection is facilitated by pressure inside the head, which can reach[38] around 100 atmospheres. The order in which crassvirus proteins are extruded through the tail is likely to be controlled both by the way these proteins are packed and by their interactions with one another. Once the cargo proteins forming the transmembrane tunnel have exited the virion, DNA could begin to descend through the tail barrel. DNA ejection could be facilitated by solid-to-fluid–like transitions as suggested for bacteriophage λ[39,40].

When considering the roles of the cargo proteins, it should be noted that members of the epsilon group of *Crassvirales* lack a detectable homologue of gp45, whereas homologues of gp46 are missing in the *Suoliviridae* (delta) and zeta groups (Fig. 1d). Thus, the sets of cargo proteins and accordingly the ejection mechanisms may vary across the crassviruses.

In conclusion, the reconstruction of the ΦcrAss001 virion identifies the structural hallmarks of crassviruses and—combined with comparative genomic analysis—has enabled us to assign functions to the majority of the previously uncharacterized proteins involved in virion assembly and infection. Among these structural features, the muzzle protein containing the novel crass fold functions as a gatekeeper, which probably contributes to the contact between the virion and the host cell surface. The extended tail barrel formed by widely conserved paralogous ring proteins has a high capacity for storage of cargo proteins alongside the capsid cargo zone, ensuring that the proteins required for the initial stages of crassvirus infection are carried inside virions and delivered to the host cell. The presence of a cargo protein storage zone in the tail is a distinct feature that, to our knowledge, has not been observed in viruses before, and suggests a mechanism of compartmentalization of the virus genome and cargo proteins ensuring their ordered release into the host.

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

## Methods

### Virus propagation and purification

The host strain *B. intestinalis* APC919/174 was propagated anaerobically in Fastidious Anaerobe Broth (Neogen) at 37 °C. Cultures were infected once a value of $A_{600}$ = 0.2 (~2 × 10$^8$ colony-forming units per ml) was reached, at a multiplicity of infection (MOI) of 1, and grown overnight at 37 °C in anaerobic jars. Cultures were centrifuged for 15 min at 5,200g at 4 °C and filtered through 0.45-μm polyethersulfone (PES) filters. Filtered lysate was concentrated by polyethylene glycol precipitation by addition of solid PEG-8000 to 10% w/v and NaCl to 0.5 M and incubation overnight at 4 °C. The solution was centrifuged at 5,000g for 30 min at 4 °C, and the pellet resuspended in a minimum volume of SM buffer (100 mM NaCl, 10 mM MgSO$_4$, 50 mM Tris-HCl pH 7.5). The solution was applied to the top of a caesium chloride step gradient in an Ultra-Clear tube (Beckman), comprising of solutions of densities ~1.31 and ~1.55 g ml$^{-1}$ in 20 mM Tris-HCl pH 7.5, 10 mM MgSO$_4$. Gradients were centrifuged at 28,000 rpm in an SW28 rotor at 4 °C for 3 h. The band containing mature virions was extracted by needle side-puncture. Samples were dialysed against buffer containing 20 mM Tris-HCl pH 7.5, 150 mM NaCl, 10 mM MgSO$_4$ overnight at 4 °C. Virus particles were further concentrated by centrifugation at 40,000 rpm in an SW41 Ti rotor at 8 °C for 1 h and resuspending in 50 μl of buffer. This sample was used for all subsequent electron microscopy analysis.

### Electron microscopy

For negative staining, carbon-formvar coated copper grids (Agar Scientific) were plasma cleaned in a PELCO easiGlow for 60 s at 0.38 mbar (air) and 20 mA. Sample solution (5 μl) was applied to a grid for 1 min then blotted with filter paper, and stained with 5 μl of 2% w/v uranyl acetate. Micrographs were recorded using a Tecnai 12 G2 BioTWIN microscope with tungsten filament operating at 120 kV, with a Ceta 16M camera (Thermo Fisher Scientific). For initial screening of samples, 16 negatively stained micrographs were recorded at magnifications of 9,300–98,000×. For cryo-EM, sample was applied to lacey carbon grids supporting an ultrathin carbon layer on 400-mesh copper (Ted Pella), glow discharged as above, and vitrified using a Vitrobot Mark IV (Thermo Fisher Scientific) at 100% humidity and 4 °C. Data were collected on a 300 kV Titan Krios (Thermo Fisher Scientific) with images recorded on a Falcon3EC detector operating in integrating mode, using EPU software (2.5.0). In total, four datasets were recorded (Extended Data Table 1) and processed together in RELION 3.1[41]. Motion correction was performed using RELION and contrast transfer functions (CTFs) were estimated using CTFFIND4[42] using dose-weighting. Empty micrographs containing no virus particles were excluded from processing beyond CTF estimation. Particles selected after 2D classification were used for 3D refinements, initially using icosahedral symmetrical averaging. To check if magnified pixel size calibrations were consistent across datasets, reconstructions were compared using determine_relative_pixel_size.py (890eb35)[43] and by manually inspection, and pixel sizes scaled to reference optics group 1. A symmetry-expanded set of particles was generated using relion_particle_symmetry_expand. The unique vertex of the capsid was identified by application of a cylindrical mask on one vertex and 3D classification into 10 classes without performing image alignment. Particles with tails inside the masked region after classification were selected for further rounds of 3D refinement using local searches, after extracting sub-particles using relion_localized_reconstruction[44]. Alternatively, in the case of the muzzle assembly, sub-particles were extracted prior to masked classification. Additional symmetry expansions were performed to resolve the muzzle assembly from tail barrel particles refined with $C_{12}$ symmetry, and of the asymmetric portal-containing vertex. Refinements were followed by CTF refinement of per-particle defocus values and astigmatism, and magnification anisotropy estimation. For the symmetry-free reconstruction of the whole virion, re-extracted particles were enlarged to include the whole of the virus particle images centred on the virion and reconstructed with Ewald sphere correction. Resolutions of the reconstructions were estimated using the Fourier shell correlation 0.143 threshold and using RELION's local resolution estimation. Owing to the large particle size, gold diffraction data collected for optics group 1 was used to check pixel size calibration and magnification anisotropy[45], and the final pixel size scaled during post-processing. Atomic models (Extended Data Fig. 6) were generated by manual building in Coot (0.9.8.1)[46], followed by real space refinement in Phenix (1.19)[47]. Images were generated using ChimeraX (1.5)[48].

### Mass spectrometry analysis

Two samples of purified virions were analysed: one sample, R, was reduced and the other, NR, was untreated before running on 4–12% SDS–PAGE (Invitrogen), and the Coomassie-stained band was excised. Sample R was further reduced with 1.5 mg ml$^{-1}$ dithioerythritol in-gel, while NR was untreated. Both samples were alkylated with 9.5 mg ml$^{-1}$ iodoacetamide and digested overnight with trypsin. Resulting peptides were analysed by liquid chromatography–tandem mass spectrometry (LC–MS/MS) over 1 h acquisitions with elution from a 50 cm EasyNano PepMap column into an Orbitrap Fusion Tribrid mass spectrometer operated in DDA mode. Duplicate acquisitions were performed for each sample, one with MS2 measured in the liner ion trap and the other with high resolution MS2 acquired in the Orbitrap mass analyser. Spectra were peak picked and searched against a sequence database containing ΦcrAss001 hypothetical proteins and common proteomic contaminants. Oxidation of methionine and carbamidomethylation of cystine were set as variable modifications. Identified peptides were filtered to 1% false discovery rate as assessed against a reversed database search. Relative quantification between samples and acquisitions was performed within PEAKS Studio (10.5)[49] from areas of aligned extracted ion chromatograms. The mean of the peak areas across the ion trap and Orbitrap acquisitions were calculated for each protein in R and NR samples respectively.

### AlphaFold structure predictions

Structure predictions using AlphaFold2[37] were performed for ΦcrAss001 protein sequences: gp46, gp47, gp49, gp50, gp21 and gp29. In the case of gp21 and gp29, the C-terminal domains that could not be modelled ab initio using electron microscopy reconstructions (owing to limited resolution and flexibility in these parts) were rigid-body docked into the density maps above the corresponding gp21 and gp29 capsid binding sites. Complete pseudo-atomic models for each fibre protein oligomer were created by fusing the N-terminal (cryo-EM modelled) and C-terminal (AlphaFold modelled) regions linked in coot.

### Protein sequence analysis

To calculate the representation of ΦcrAss001 virion protein homologues across the *Crassvirales*, the protein sequences encoded in the previously described reference genome set[5] were searched. The reference set consisted of 673 virus genomes grouped into 211 clusters at 90% similarity at the genome nucleotide sequence level. Each virion protein was queried against the reference set using PSI-BLAST[50]. To account for over-representation of some crassviruses, a genome cluster was given a score of 1 if a ΦcrAss001 protein homologue was found in more than half of the genomes in that cluster (otherwise 0). The cluster scores were then summed within the previously delineated groups of crassviruses (alpha to zeta) and plotted against the corresponding ΦcrAss001 proteins (Extended Data Table 4, Fig. 1d). To identify homologues outside the *Crassvirales*, the Genbank and RefSeq virus databases were queried using PSI-BLAST.

### Reporting summary

Further information on research design is available in the Nature Portfolio Reporting Summary linked to this article.

## Data availability

Cryo-EM reconstructions and atomic coordinates comprising the virion capsid and tail structures have been deposited to the Electron Microscopy Data Bank under accession codes EMD-14088, EMD-14089, EMD-14090, EMD-14091, EMD-14092, EMD-14093, EMD-14094 and EMD-14100 (maps) and to the Protein Data Bank under accession codes 7QOF, 7QOG, 7QOH, 7QOI, 7QOJ, 7QOK and 7QOL (atomic coordinates), with example regions of maps and corresponding models in Extended Data Fig. 6. Data collection and refinement statistics are presented in Extended Data Table 1. Multiple sequence alignments of ΦcrAss001 virion protein homologues are available at https://ftp.ncbi.nih.gov/pub/yutinn/crAss_structural_prot_2022/.

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

**Acknowledgements** The authors thank S. Hart and J. Turkenburg for assistance with cryo-EM; A. Dowle and C. Taylor for assistance with mass spectrometry experiments; R. Thompson, E. Hesketh and D. Maskell; and support staff at eBIC (Diamond Light Source, UK). This project used the Viking Cluster high performance compute facility provided by the University of York; we are grateful for computational support from the University of York High Performance Computing service, Viking and the Research Computing team. This work was supported by the Wellcome Trust (grant 206377, 224665 to A.A.A.), Instruct-ERIC (PID: 10175), and Diamond Light Source for access to eBIC under proposal EM19832 funded by the Wellcome Trust, MRC and BBRSC. The cryo-EM facility at the University of York was supported by the Wellcome Trust, grant 206161. A.N.S. was supported by the Wellcome Trust (grant 220646) and ERC (grant agreement no. 101001684). E. V. Khokhlova and C.H. were supported by Science Foundation Ireland (SFI) under Grant no. SFI/12/RC/2273. N.Y. and E. V. Koonin are supported by the Intramural Research Program of the National Institutes of Health of the USA (National Library of Medicine). Molecular graphics and analyses were performed with UCSF ChimeraX, developed by the Resource for Biocomputing, Visualization, and Informatics at the University of California, San Francisco, with support from National Institutes of Health R01-GM129325 and the Office of Cyber Infrastructure and Computational Biology, National Institute of Allergy and Infectious Diseases.

**Author contributions** O.W.B., A.N.S., E. V. Koonin, C.H. and A.A.A. conceived the study. A.N.S. and E. V. Khokhlova cultured cells and viruses. O.W.B., A.N.S. and E. V. Khokhlova purified virions. O.W.B. and D.E.D.P.H. collected cryo-EM data. O.W.B. processed cryo-EM data. O.W.B. and J.L.R.S. built and refined atomic models. O.W.B. and A.A.A. analysed structural data. N.Y. and E. V. Koonin performed comparative genomic analysis. O.W.B., A.N.S., E. V. Koonin, C.H. and A.A.A. wrote the manuscript, which was read, edited and approved by all authors.

**Competing interests** The authors declare no competing interests.

**Additional information**
**Correspondence and requests for materials** should be addressed to Oliver W. Bayfield or Alfred A. Antson.

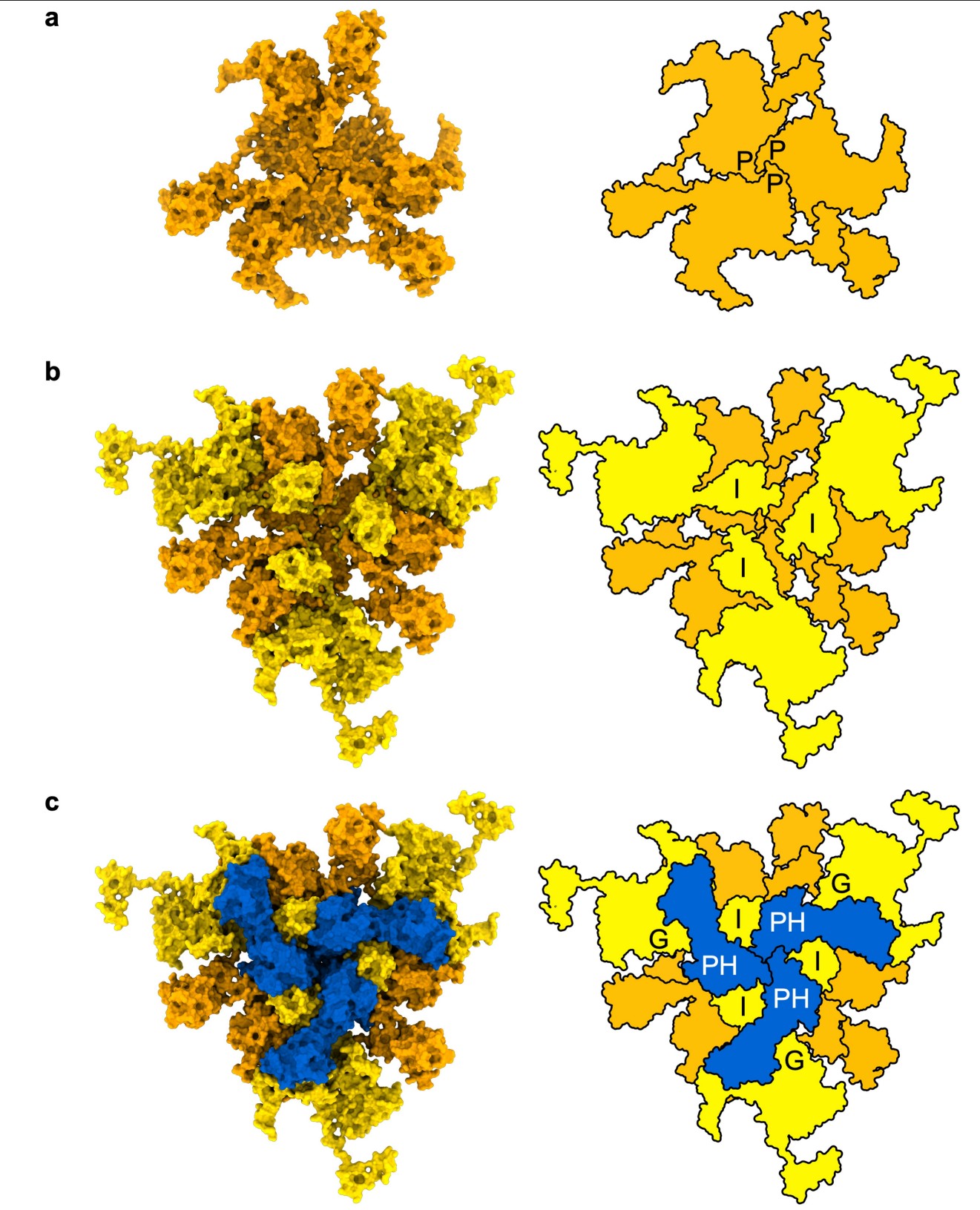

**Extended Data Fig. 1 | Capsid wall viewed along the local three-fold symmetry axis.** Molecular surface (left) and diagram of protein layers (right). **(a)** Inner capsid layer formed by three major capsid protein subunits contributed from three adjacent capsomers, orange; **(b)** Middle layer containing three additional overlaying major capsid protein subunits, yellow; **(c)** Outer layer containing three overlaying auxiliary capsid proteins, blue. P, peripheral domain of the major capsid protein. I, insertion domain of the major capsid protein. PH, pleckstrin-homology domain of the auxiliary capsid protein. G, G-loop of the major capsid protein.

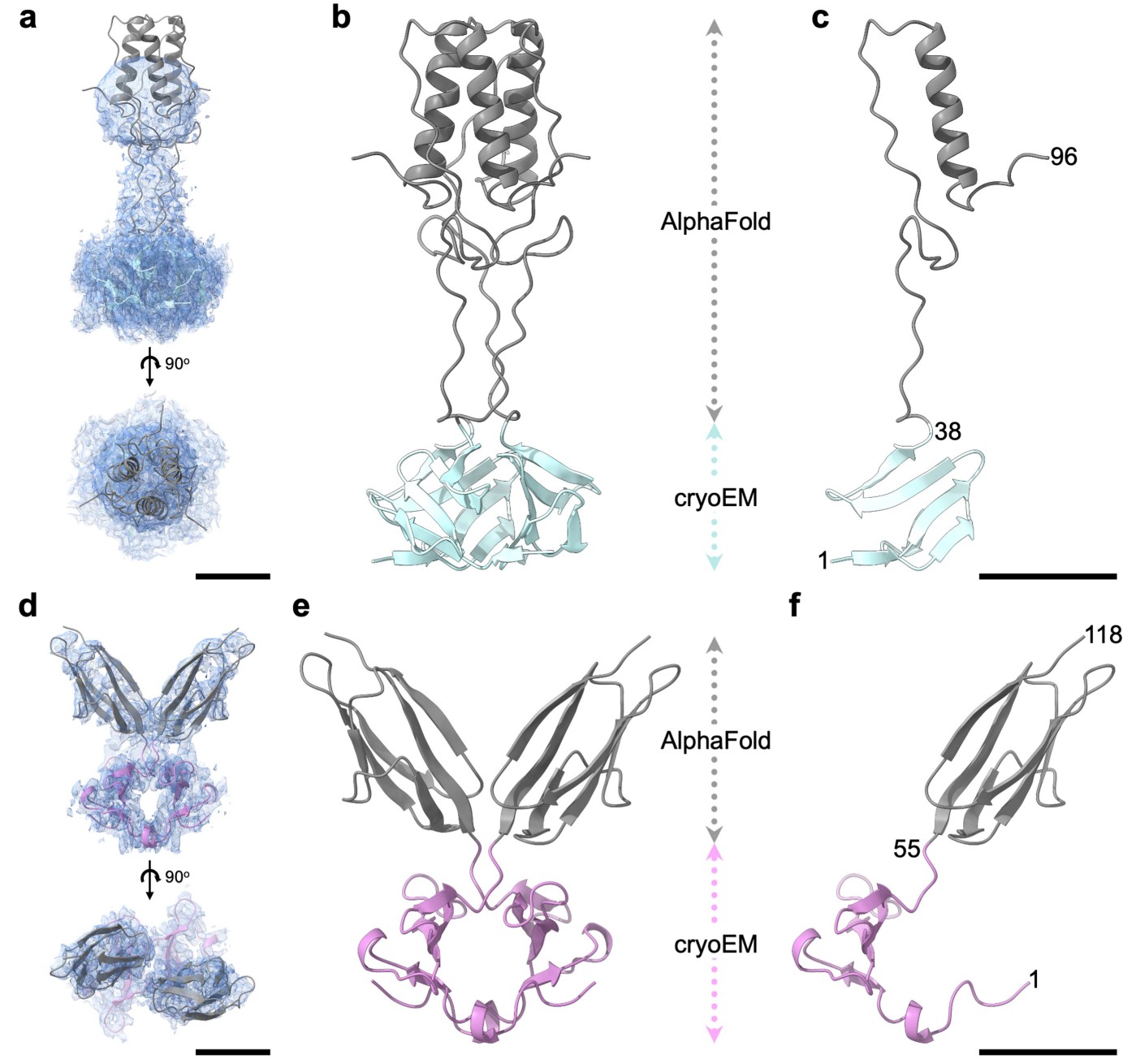

**Extended Data Fig. 2 | Models of the head fiber proteins. (a–c)**, head fiber gp21 trimer. **(d–f)**, head fiber gp29 dimer. **(a)** Extracted density fitted with the gp21 model; shown in two orthogonal views. **(b)** Ribbon diagram with domain structure deduced from cryo-EM density in blue and domain structure predicted by AlphaFold in grey. **(c)** same as B but with only one monomer shown. **(d–f)** same as (a-c) but for gp29, with the domain structure derived from cryo-EM density shown in pink. Scale bars 20 Å.

MAKKKIKRRGKMPPNIFDTGGQSWGQQSSGQFSNAFKGENLGNSIGSIGGAVGGIAQAG
ISNAQIADTSGIEAQNKAQKNMVVGASSNDDLMSEWGSWNKVKDDYSWKDVRGGSTGQR
VTNTIGAAGQGAAAGASVGGPIGAIVGGVVGLGSAIGGWLGGNRKAKRKAKKLNKEAKE
ANERALTSFETRADNIDTQNDFNMLANFSAYGGPLE**FGSGAIGYEFDNRYLNNQEMSAV
AKQRLTSLP**NSFQALPEMNTYNAFAEGGGLSREKNYGSKKKPYPSVPSGDFAGPHRSYP
IPTKADARDALRLAGLHGNESVRRKVLAKYPSLKAFGGSLFDSVVGNNFNQSFTQGIQG
MFQQEPEQTVQAANIAKDGGDIKIKEKNKGKFTAYCGGKVTEACIRKGKNSSNPTTRKR
ATFAQNARNWN**AFGGWLNTQGGDFTNGVTFINEGGSHEENPYQGIQIGVDPEGAPNLVE
QGEVVYDDYVFSD**RMEIPDDIRKE**YKLRGKTFAKAAKSAQRESEERPNDPLSTKGLQAA
MERI**ATAQEEARQRKEAHREGNEYPSMFAYGGDTNPYGLALEDPMSVEELEALMVQSGE
TGEIAPEGNNGNRQTWTRYAPIIGSGLASLSDLFSKPDYDSADLISGVDLGAEAVGYAP
IGNYLSYRPLDRDFYINKMNQQAAATRRGLMNTSGGNRLNAQAGILAADYNYGQNMGNL
ARQAEEYNQQLRERVEAFNRGTNMFNTETGLKASMFNAESRNAAKRARLGQATTVAQLR
QGIKDQDAARRSANITNFLQGLGDMGWENEQANWLDTLAKSGVLKMNTKGEYTGGTKKA
KGGKVRTKKKKGLTYG

**Extended Data Fig. 3 | Sequence of the cargo protein gp45.** Regions detected by mass spectrometry are in purple, structured regions identified in cryo-EM density are bold and underlined; the predicted transmembrane helix is in yellow.

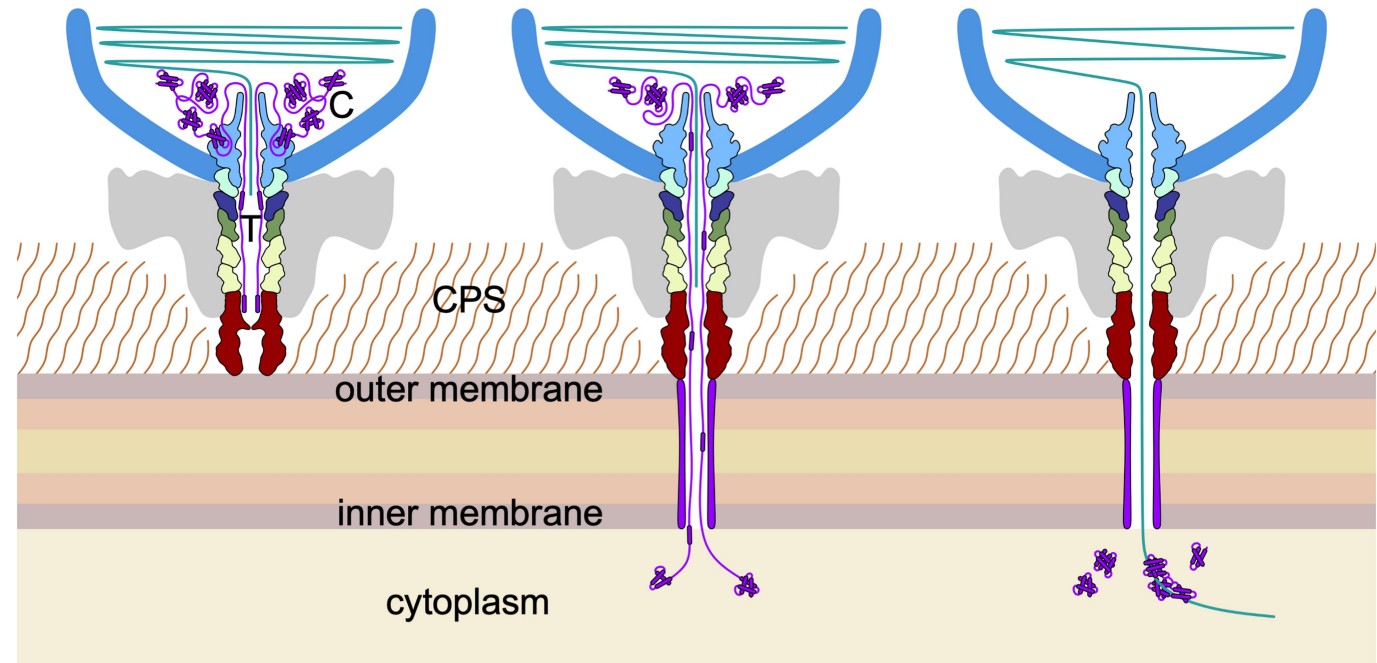

**Extended Data Fig. 4 | Model for ejection of cargo proteins.** Left cargo proteins (purple) are stored in the tail barrel and capsid cargo zones in association with DNA (green). Middle: cargo proteins are being extruded through the constrictions of the portal and tail barrel and are passing through the transmembrane channel (purple) into the host's cytoplasm. Injected proteins can undergo refolding in the cytoplasm. Right: cargo proteins' ejection is complete and is followed by DNA ejection. Viral RNA polymerases have re-folded and transcription of viral genome begins. CPS, capsular polysaccharide.

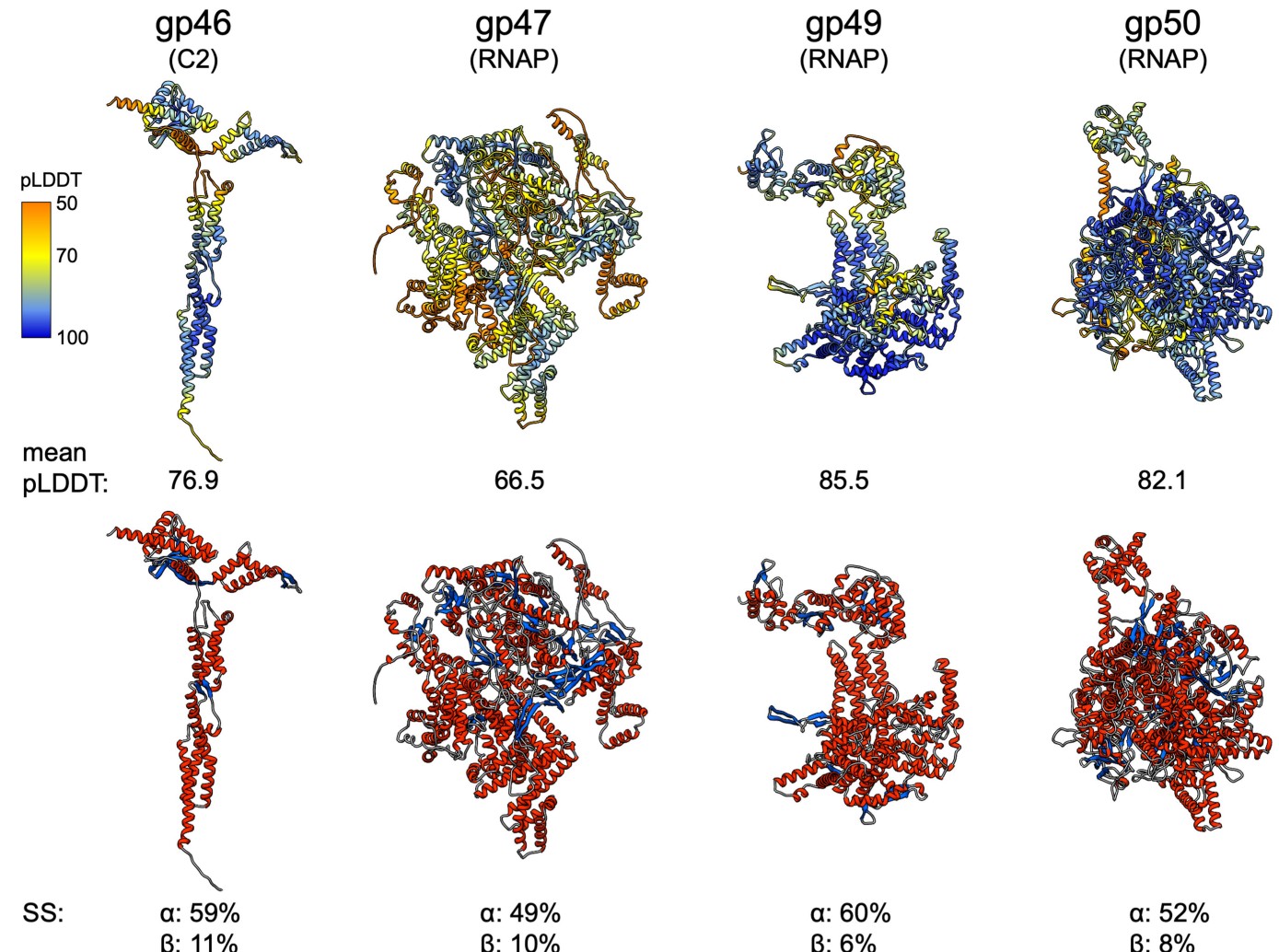

**Extended Data Fig. 5 | Ribbon diagrams of cargo protein structures predicted by AlphaFold.** Top row, coloured according to the pLDDT score: 50, orange; 70, yellow; 100, blue. Mean overall pLDDT is indicated underneath each protein structure. Bottom row, coloured according to secondary structure (α-helix, red; β-strand, blue; coil, grey) with overall percentages of α- and β- residues indicated underneath.

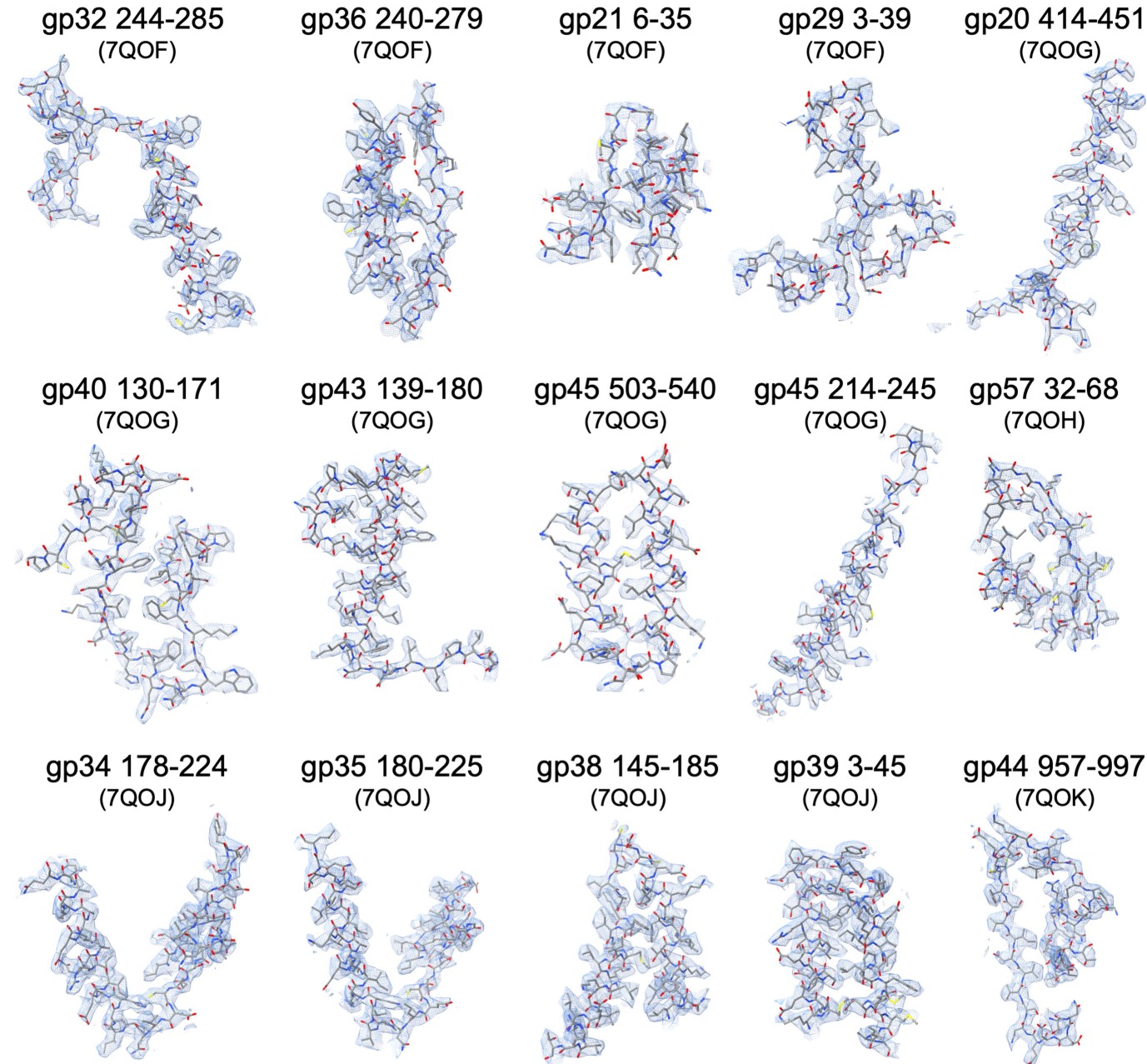

**Extended Data Fig. 6 | Representative density maps with corresponding fitted models for regions of virion proteins.** Maps are depicted in blue mesh with atomic models shown in stick style coloured by atom type. Each protein is labelled with residue ranges shown and the corresponding PDB codes indicated in parentheses.

**Extended Data Table 1 | Data collection and refinement**

**Data collection**

| Dataset (optics group) | 1 | 2 | 3 | 4 |
|---|---|---|---|---|
| Voltage / kV | 300 | 300 | 300 | 300 |
| Magnification | 75,000 | 59,000 | 59,000 | 59,000 |
| Defocus range / μm | 0.4–2.0 | 0.5–1.5 | 0.3–1.5 | 0.3–1.5 |
| No. micrographs | 7506 | 6515 | 11593 | 18392 |
| Pixel size / Å | 1.065 | 1.400 | 1.390 | 1.390 |
| Relative pixel size / Å | 1.065 | 1.397 | 1.370 | 1.368 |
| Dose rate / pixel sec$^{-1}$ | 54.16 | 70.36 | 86.05 | 63.00 |
| Exposure time / sec | 1.00 | 1.40 | 1.17 | 1.55 |
| No. fractions | 39 | 46 | 45 | 45 |

**Data processing and Refinement**

| Assembly | Capsid (EMD-14088) (PDB 7QOF) | Portal (EMD-14089) (PDB 7QOG) | Vertex C5 (EMD-14090) (PDB 7QOH) | Vertex C1 (EMD-14091) (PDB 7QOI) | Barrel (EMD-14092) (PDB 7QOJ) | Muzzle (EMD-14093) (PDB 7QOK) | Tail (EMD-14094) (PDB 7QOL) | Virion (EMD-14100) |
|---|---|---|---|---|---|---|---|---|
| Symmetry imposed | I | $C_{12}$ | $C_5$ | $C_1$ | $C_{12}$ | $C_6$ | $C_6$ | $C_1$ |
| Pixel size / Å | 1.365 | 1.394 | 1.394 | 1.394 | 1.394 | 1.394 | 1.394 | 1.999 |
| No. particles | 55,063 | 122,709 | 122,709 | 122,709 | 122,709 | 122,709 | 46,280 | 27,445 |
| Map resolution, FSC 0.143 / Å | 3.01 | 3.09 | 3.32 | 3.62 | 3.21 | 3.38 | 3.33 | 4.40 |
| Map sharpening B-factor / Å$^2$ | -152 | -171.9 | -156.5 | -159.7 | -168.5 | -179.9 | -129 | -102 |
| Map resolution range / Å | ~2.8–5.2 | ~2.9–8.9 | ~2.8–8.1 | ~3.3–9.7 | ~2.9–12.2 | ~3.2–13.9 | ~3.1–9.8 | ~4.0–38.8 |
| Map-Model correlation | 0.8338 | 0.8408 | 0.8176 | 0.7861 | 0.798 | 0.8223 | 0.834 | - |
| Model composition | | | | | | | | |
| Non-hydrogen atoms | 59,983 | 10,291 | 37,081 | 310,243 | 20,218 | 14,808 | 51,480 | - |
| Protein residues | 7647 | 1266 | 4705 | 38,881 | 2497 | 1851 | 6371 | - |
| Ligands | 1 | 0 | 6 | 27 | 1 | 1 | 2 | - |
| R.M.S. deviations | | | | | | | | |
| Bond lengths / Å | 0.005 | 0.008 | 0.005 | 0.007 | 0.006 | 0.004 | 0.004 | - |
| Bond angles / ° | 0.613 | 0.881 | 0.790 | 0.905 | 0.755 | 0.707 | 0.691 | - |
| Ramachandran plot | | | | | | | | |
| Favoured / % | 95.00 | 97.11 | 96.09 | 95.80 | 96.66 | 95.71 | 96.69 | - |
| Allowed / % | 4.86 | 2.89 | 3.80 | 4.17 | 3.30 | 4.29 | 3.31 | - |
| Outlier / % | 0.14 | 0.00 | 0.11 | 0.02 | 0.04 | 0.00 | 0.00 | - |
| Validation | | | | | | | | |
| Rotamer outliers / % | 0.02 | 0.27 | 0.25 | 0.99 | 0.22 | 0.55 | 0.19 | - |
| Clashscore | 7.25 | 4.74 | 4.02 | 5.71 | 3.50 | 3.88 | 3.78 | - |
| Molprobity | 1.75 | 1.40 | 1.45 | 1.60 | 1.35 | 1.47 | 1.38 | - |

Data collection parameters are shown for different data sets, with refinement statistics shown for different assemblies.

**Extended Data Table 2 | Disulfide bonds stabilising the tail assembly**

| Interaction (Residue 1–Residue 2) | | | | Confidence | Notes |
|---|---|---|---|---|---|
| Hub-Hub | | | | | |
| gp39(THB) | 65 | 163 | gp38(THA1) | +++ | gp38 Cys163 can bond to either gp39 Cys65 (HA1 to HB) or gp43 Cys66 (HA2 to R1). |
| gp38(THA1) | 131 | 211 | gp38(THA2) | +++ | gp38 Cys131 can bond to either gp38 Cys211 (HA1 to HA2) or gp38 Cys213 (HA2 to HA1). |
| gp38(THA2) | 131 | 213 | gp38(THA2) | +++ | gp38 Cys131 can bond to either gp38 Cys211 (HA1 to HA2) or gp38 Cys213 (HA2 to HA1). |
| Hub-Ring | | | | | |
| gp43(R1) | 66 | 163 | gp38(THA2) | +++ | gp38 Cys163 can bond to either gp39 Cys65 (HA1 to HB) or gp43 Cys66 (HA2 to R1). |
| gp39(THB) | 71 | 74 | gp40(R2) | +++ | |
| Intramolecular | | | | | |
| gp38(THA2) | 111 | 204 | gp38(THA2) | +++ | |
| gp40(R2) | 60 | 170 | gp40(R2) | ++ | |
| Ring-Ring | | | | | |
| gp35(R3) | 116 | 178 | gp40(R2) | + | |

Residues forming disulfide interactions stabilising protein-protein interactions are indicated by numbers.

**Extended Data Table 3 | Virion proteins detected by mass spectrometry**

| Gp | Description | No. unique peptides | Coverage /% | R Area | NR Area |
|----|-------------|---------------------|-------------|--------|---------|
| gp20 | Portal protein | 35 | 57 | 3.30E+08 | 3.01E+07 |
| gp21 | Head fiber trimer protein | 5 | 72 | 2.60E+07 | 6.98E+06 |
| gp22 | Tail fiber protein | 39 | 62 | 8.26E+08 | 5.62E+08 |
| gp23 | Tail fiber protein | 32 | 53 | 5.14E+08 | 4.99E+08 |
| gp24 | Putative C-type lectin | 19 | 71 | 4.95E+08 | 4.41E+08 |
| gp25 | Tail fiber protein | 38 | 61 | 8.34E+08 | 1.48E+09 |
| gp26 | Tail fiber protein | 43 | 57 | 8.81E+08 | 1.48E+09 |
| gp29 | Head fiber dimer protein | 9 | 85 | 3.19E+08 | 3.97E+07 |
| gp32 | Major capsid protein | 41 | 74 | 2.34E+10 | 1.61E+09 |
| gp36 | Auxiliary capsid protein | 20 | 56 | 1.34E+10 | 8.65E+08 |
| gp43 | Ring Protein 1 | 6 | 30 | 8.02E+07 | 4.29E+05 |
| gp44 | Muzzle protein | 25 | 25 | 4.55E+07 | 2.29E+06 |
| gp45 | Cargo protein 1 | 16 | 24 | 1.24E+08 | 1.49E+07 |
| gp46 | Cargo protein 2 | 30 | 80 | 3.36E+08 | 4.28E+07 |
| gp47 | RNA polymerase subunit | 97 | 53 | 5.85E+08 | 2.24E+07 |
| gp49 | RNA polymerase subunit | 67 | 47 | 3.56E+08 | 1.29E+07 |
| gp50 | RNA polymerase subunit | 113 | 51 | 8.02E+08 | 2.12E+07 |

Individual proteins are shown with the number of unique peptides detected and the overall sequence coverage.

**Extended Data Table 4 | Representation of ΦcrAss001 protein homologues across crassviruses**

|  | POR | MCP | R3/4 | R2 | R1 | MUZ | C1 | C2 | AUX | THA | THB | HFT | HFD | PVA |
|---|---|---|---|---|---|---|---|---|---|---|---|---|---|---|
| *Intestiviridae* | 1.00 | 1.00 | 1.00 | 1.00 | 1.00 | 1.00 | 1.00 | 1.00 | 0.83 | 0.00 | 0.00 | 0.35 | 0.00 | 0.00 |
| *Crevaviridae* | 1.00 | 1.00 | 1.00 | 1.00 | 1.00 | 1.00 | 1.00 | 1.00 | 0.25 | 0.00 | 0.00 | 0.00 | 0.26 | 0.00 |
| *Steigviridae* | 1.00 | 1.00 | 1.00 | 1.00 | 1.00 | 1.00 | 1.00 | 1.00 | 1.00 | 0.90 | 0.93 | 0.72 | 0.24 | 0.34 |
| *Suoliviridae* | 1.00 | 1.00 | 0.95 | 0.95 | 0.95 | 0.97 | 0.97 | 0.00 | 0.95 | 0.01 | 0.01 | 0.17 | 0.00 | 0.00 |
| zeta | 1.00 | 1.00 | 0.97 | 0.97 | 0.97 | 0.94 | 0.88 | 0.00 | 0.06 | 0.03 | 0.03 | 0.00 | 0.00 | 0.00 |
| epsilon | 1.00 | 1.00 | 1.00 | 1.00 | 1.00 | 1.00 | 0.26 | 1.00 | 0.00 | 0.00 | 0.00 | 0.00 | 0.00 | 0.00 |
| ProJPt-Bp1 | 1.00 | 1.00 | 1.00 | 1.00 | 1.00 | 1.00 | 1.00 | 0.00 | 1.00 | 0.00 | 0.00 | 0.00 | 0.00 | 0.00 |
| Fpv3 | 1.00 | 1.00 | 1.00 | 1.00 | 1.00 | 1.00 | 1.00 | 1.00 | 1.00 | 0.00 | 0.00 | 0.00 | 0.00 | 0.00 |
| Φ14:2 | 1.00 | 1.00 | 1.00 | 1.00 | 1.00 | 1.00 | 1.00 | 1.00 | 1.00 | 0.00 | 0.00 | 0.00 | 0.00 | 0.00 |
| Φ17:2 | 1.00 | 1.00 | 1.00 | 1.00 | 1.00 | 1.00 | 1.00 | 1.00 | 0.00 | 0.00 | 0.00 | 0.00 | 0.00 | 0.00 |
| Φ13:2 | 1.00 | 1.00 | 1.00 | 0.00 | 0.00 | 1.00 | 0.00 | 0.00 | 0.00 | 0.00 | 0.00 | 0.00 | 0.00 | 0.00 |

Scores shown were calculated as described in Methods.

# Reporting Summary

## Statistics

For all statistical analyses, confirm that the following items are present in the figure legend, table legend, main text, or Methods section.

| n/a | Confirmed | |
|---|---|---|
| ☐ | ☒ | The exact sample size (*n*) for each experimental group/condition, given as a discrete number and unit of measurement |
| ☐ | ☒ | A statement on whether measurements were taken from distinct samples or whether the same sample was measured repeatedly |
| ☒ | ☐ | The statistical test(s) used AND whether they are one- or two-sided *Only common tests should be described solely by name; describe more complex techniques in the Methods section.* |
| ☒ | ☐ | A description of all covariates tested |
| ☒ | ☐ | A description of any assumptions or corrections, such as tests of normality and adjustment for multiple comparisons |
| ☒ | ☐ | A full description of the statistical parameters including central tendency (e.g. means) or other basic estimates (e.g. regression coefficient) AND variation (e.g. standard deviation) or associated estimates of uncertainty (e.g. confidence intervals) |
| ☒ | ☐ | For null hypothesis testing, the test statistic (e.g. *F*, *t*, *r*) with confidence intervals, effect sizes, degrees of freedom and *P* value noted *Give P values as exact values whenever suitable.* |
| ☒ | ☐ | For Bayesian analysis, information on the choice of priors and Markov chain Monte Carlo settings |
| ☒ | ☐ | For hierarchical and complex designs, identification of the appropriate level for tests and full reporting of outcomes |
| ☒ | ☐ | Estimates of effect sizes (e.g. Cohen's *d*, Pearson's *r*), indicating how they were calculated |

*Our web collection on statistics for biologists contains articles on many of the points above.*

## Software and code

Policy information about availability of computer code

| Data collection | EPU 2.5.0 |
|---|---|
| Data analysis | RELION 3.1; CTFFIND4; Phenix 1.19; Coot 0.9.8.1; ChimeraX 1.5; determine_relative_pixel_size.py 890eb35; TMHMM-2.0; PEAKS Studio 10.5. |

For manuscripts utilizing custom algorithms or software that are central to the research but not yet described in published literature, software must be made available to editors and reviewers. We strongly encourage code deposition in a community repository (e.g. GitHub). See the Nature Portfolio guidelines for submitting code & software for further information.

## Data

Policy information about availability of data

All manuscripts must include a data availability statement. This statement should provide the following information, where applicable:

- Accession codes, unique identifiers, or web links for publicly available datasets
- A description of any restrictions on data availability
- For clinical datasets or third party data, please ensure that the statement adheres to our policy

Cryo-EM reconstructions and atomic coordinates comprising of the virion capsid and tail structures have been deposited with wwPDB (www.wwpdb.org) under accession codes: EMD-14088, 14089, 14090, 14091, 14092, 14093, 14094, 14100 (maps) and 7QOF, 7QOG, 7QOH, 7QOI, 7QOJ, 7QOK, 7QOL (atomic coordinates), with examples regions of maps and corresponding models in Extended Data Figure 6. Data collection and refinement statistics are presented in Extended Data Table 1. Multiple sequence alignments of ΦcrAss001 virion protein homologues are available at https://ftp.ncbi.nih.gov/pub/yutinn/crAss_structural_prot_2022/.

## Human research participants

Policy information about studies involving human research participants and Sex and Gender in Research.

| | |
|---|---|
| Reporting on sex and gender | N/A as no human research participants. |
| Population characteristics | N/A as no human research participants. |
| Recruitment | N/A as no human research participants. |
| Ethics oversight | N/A as no human research participants. |

Note that full information on the approval of the study protocol must also be provided in the manuscript.

# Field-specific reporting

Please select the one below that is the best fit for your research. If you are not sure, read the appropriate sections before making your selection.

☒ Life sciences          ☐ Behavioural & social sciences          ☐ Ecological, evolutionary & environmental sciences

For a reference copy of the document with all sections, see nature.com/documents/nr-reporting-summary-flat.pdf

# Life sciences study design

All studies must disclose on these points even when the disclosure is negative.

| | |
|---|---|
| Sample size | A total of 122,709 images of individual virus particles were used in reconstructions of the virions and sub-assemblies. |
| Data exclusions | Empty micrographs containing no virus particles were excluded. Images of virus particles were extracted and classified according to cryo-EM data processing procedures in RELION. |
| Replication | A single sample of purified virus particles derived from bacterial lysate was used to prepare electron microscopy grids. Four cryo-EM data sets of micrographs of virus particles were processed, in total comprising 44,006 micrographs. In addition, 16 negatively stained micrographs were used for initial assessments of sample quality. |
| Randomization | A bacterial colony or viral plaque was selected at random to initiate propagation of cultures (each colony or plaque contains 1000–10,000 bacteria or viruses). A random subset of virus particles present in solution was embedded in vitreous ice upon freezing. |
| Blinding | No blinding was performed as the subjects were viruses and bacteria. Cryo-EM data were randomly divided into half-sets and refined independently according to the gold-standard procedure implemented in RELION. |

# Reporting for specific materials, systems and methods

We require information from authors about some types of materials, experimental systems and methods used in many studies. Here, indicate whether each material, system or method listed is relevant to your study. If you are not sure if a list item applies to your research, read the appropriate section before selecting a response.

#### Materials & experimental systems

| n/a | Involved in the study |
|---|---|
| ☒ ☐ | Antibodies |
| ☒ ☐ | Eukaryotic cell lines |
| ☒ ☐ | Palaeontology and archaeology |
| ☒ ☐ | Animals and other organisms |
| ☒ ☐ | Clinical data |
| ☒ ☐ | Dual use research of concern |

#### Methods

| n/a | Involved in the study |
|---|---|
| ☒ ☐ | ChIP-seq |
| ☒ ☐ | Flow cytometry |
| ☒ ☐ | MRI-based neuroimaging |

