## [Peer Review File · Nature]

Manuscript Title: Structural atlas of a human gut crAssvirus

Reviewer Comments & Author Rebuttals

Reviewer Reports on the Initial Version:

Referee #1 (Remarks to the Author):

The manuscript by Bayfield et. al., present a cryo-electron microscopy analysis and structural reconstruction of the bacteriophage crAssphage, which is the most abundant virus within the human gut. The importance of crAssphage has recently been highlighted through its dominance of the gut, accounting for upwards of 90% of the virions in certain individuals. However, the biological relevance, function and inner workings of this virus remain largely unresolved. This manuscript provides a high-resolution analysis of the structural components of the crAssphage virion and blends these with a number of unique observations and biological inferences.

I should note that I am not a structural biologist and did not critically assess these aspects of the manuscript, but rather assessed the phage biology and broader relevance of the work. I have no doubts this work will be important to the structural biology field. More broadly, I do think this work has significant impact and interest to warrant consideration by Nature. This is due to the uniqueness and dominance of this virus within the human microbiome and this manuscript makes several novel observations and insights into the structures present within this virion and their putative function. I would like to commend the authors on their writing and the incorporation of biological insights into the manuscript, particularly the sections describing the structure and subsequent function of the muzzle protein and their model for protein and DNA injection. These sections were accessible and provided greater biological insight into the function of crAssphage.

Largely this is an extremely well written, prepared, and pitched manuscript and I have no major comments. I do have two small points for the authors to consider broadening the appeal of their paper outside of the structural biology field.

Lines 37-38 – When introducing crAssphage, I would recommend the authors consider adding a few more sentences to explain some relevant points on the known crAssphage infection mechanism and lifecycles. Describing host range and some base lifestyle and infection dynamics would bring additional relevance to the later sections.

Lines 109-119 - Is there any information on this fold or closest structural homologue for these two fiber vertex proteins? This is usual for a phage to alternate these capsid display proteins. There was one brief point on potential role of these structures but compared with other sections of the manuscript this section left underdone. Including some additional analyses and biological insight into the potential function of these proteins would further improve.

Referee #2 (Remarks to the Author):

Overview

In the article, Bayfield et al. "Structural atlas of the most abundant human gut virus," the authors share the molecular reconstruction of the virion of a tailed bacteriophage representative in the order Crassvirales, a frequent but still relatively unknown group of viruses found in the human gut. The cryo-electron microscopy maps led to a 3D reconstruction of the virion near 3-angstrom resolution facilitating the molecular modeling of all the structural components of the capsid and the tail. The maps also captured the internal organization of the DNA genome and the presence of cargo proteins, which the authors corroborated using mass spectroscopy. The quality of the virion reconstruction is excellent, and the results are significant for the scientific community, as highlighted in my next paragraph. Nonetheless, I found that the interpretation of the results is strongly speculative and overlooks prior results, as pointed out in my major comments below.

Relevance

The study of the crAss001 presents the first high-resolution molecular reconstruction of a representative in the crAss-like phages group, representing an essential scientific contribution to the community. The structural reconstructions will provide valuable insight to infer the structure of the order of Crassvirales, guiding further research of these frequent but elusive human gut viruses. The molecular information of the tail proteins could guide the inference of host-phage interactions in other viruses of this critical group. Additionally, several findings might particularly interest the structural virology community. The molecular structure of the virion departs significantly from other known tailed phages, for example, the type of auxiliary protein present on the capsid, the size of the tail for a podovirus-like phage, or the intriguing presence (and ejection mechanism) of the cargo proteins. Overall, this is an excellent contribution despite the criticisms added below.

Major comments

1. The authors claim that the structure suggests "a general mechanism for protein ejection, which involves partial unfolding of proteins during their extrusion through the tail." This interpretation overlooks that the virions were extracted and dialyzed at 4°C (lines 394-404), but the virion infects the host at 37°C (lines 392-393). Previous studies (not cited in the article) have shown that the packaged DNA in other tailed phage model systems, like lambda phage infecting *E. coli*, undergoes a solid-to-fluid-like transition that facilitates infection near 37°C (Liu et al. PNAS 2014; Evilevitch, eLife 2018). Therefore, one should be cautious in making interpretations of the virion obtained at 4°C. The article should state clearly (and discuss carefully) that the virion corresponds to the structure at 4°C and that, based on prior studies, the internal state of the virion (including the cargo proteins) is likely to change near 37°C. The article should tone down, reduce the length, and remove the figure regarding the current speculation.
2. The article proposes that the muzzle at the end of the tail adopts a new fold, which the authors coined as crAss fold. However, no data regarding the lack of similarity or homology with other proteins are shared (lines 248-249). What were the tests done to arrive at this conclusion? Are there no similarities between the domains of the muzzle and other proteins? What are the closest protein structures and protein topologies known? By how much it differs? The protein fold's quantitative

and qualitative assessment should be clear if it is proposed as a new fold. Additionally, a precise figure about these findings is far more necessary in the article than the current speculative ejection mechanism figure.

3. The title states that the structure reconstructed corresponds to "the most abundant human gut virus." However, it is not evident what they mean by the most abundant virus (crAss001? the whole crAss-like group?). To my understanding, the specific virus investigated, crAss001, is not the most abundant virus in the human gut. The original crAssphage obtained from uncultured genomes in Dutilh et al. Nature Comm. 2014 (ref. 2) was several times more abundant than all other known phages then. But crAss001 shares little sequence similarity with the original crAssphage. crAss001 belongs to the "group of loosely related bacteriophages termed crAss-like bacteriophages (Yutin et al. Nat. Commun. 2018) (quoted from ref. 6). But what is the actual abundance of crAss001? Additionally, the article does not include a relevant study on human gut viruses: Benler et al. Microbiome 2021. That study investigated human gut metagenomes, identifying new abundant virus groups, such as the Flandesviridae and Quimbyviridae candidate families. These families reached similar detection frequencies as crAss-like phages despite having fewer phages in the database and displaying shorter genomes. Thus, it is not even clear that the crAss-like group is that abundant compared to other new virus groups, let alone crAss001. It is undeniable that the crAss-like phage group is a frequent, cosmopolitan, and intriguing group of human gut viruses. However, any claims regarding their abundance should be more accurate, precise, and cautious in the article.

Referee #3 (Remarks to the Author):

Bayfield et al present a structural tour de force of the *Bacteroides intestinalis* bacteriophage Φ crAss001, a prototype of the crAssviruses that are purported to play major roles in shaping the composition and functionality of the human microbiome. Φ crAss001 is the first crAssvirus to be produced in culture, allowing the structural characterization presented here.

The authors provide a remarkably detailed picture of the phage structure with the focus on the novelty of some features when compared with other phage structures. The differences include the, so called, muzzle protein rings formed at the distal end of the tail and potentially participating in the phage interaction with the host cell plasma membrane. These gene products appear to be unique to the crAssviruses. In addition, there are features of the cargo proteins that appear unusual. The auxiliary capsid proteins are novel in their tertiary structure and possibly their mode of interacting with the major capsid proteins. I am not familiar with other examples of an auxiliary protein covering nearly the entire surface of a particle. A side view in figure 2 D or E would be helpful.

The paper is high density as necessitated by the large number of structural proteins identified and their interactions with neighbors. As the paper flows it is difficult to tell which of the proteins are possibly unique to this virus and which have homologs in other bacteriophages. An example is the portal protein subunit when it is introduced on line 120. It is presented without specifically saying that the structure is homologous to those previously observed and only at line 135 where they begin

describing the stem domain and other named features of portal proteins. It would be helpful if, as each protein structure is presented, it is immediately stated that the overall fold is homologous to previously reported proteins. Better yet, in line 57 it could be stated that all of the structures reported except the muzzle proteins and the auxiliary capsid protein have structural homologs in other phages and that this paper reveals the novel insertions and deletions that are the basis for the crassvirus sequence analysis presented in figure 1 C and D.

It is difficult to fully assess the technical quality of the reconstructions from the material provided for reviewed. The authors describe some remarkably detailed features for the cargo proteins that are probably present at low occupancy. They have undoubtedly applied all of the various tools for improving local resolution in the presence of high symmetry and this has been effective in discerning subtle differences in quasi-equivalent interactions.

Below are a few points that the authors should address.

Line 82 There should be a reference to the HK97 capsid protein fold.

Line 84 Is the E-loop insertion novel or are there comparable insertions in non crass phage?

Line 86 Gp36 appears to have a different association with the capsid when compared to other auxiliary proteins. As mentioned in the text it is unusual for an auxiliary protein to cover so much of the particles. It appears to interact with the MCP in multiple places. A side view in Fig. 2 C or D would clarify this.

Line 112 Are structural homologs of the fiber proteins present in other phages?

Line 128 Should say whether the portal protein has the canonical fold and a reference to the portal protein structure should be included.

Line 138 Similar structure in P22 Tang J, etal Structure. 2011 Apr 13;19(4):496-502.

Line 144 Does this mean that no symmetry was applied. State this clearly.

Line 190 What is the sequence identity among Ring proteins 1.2.3 and 4/5?

Line 196 All rings are formed by the same gene product?

Line 200 varying number of rings in other viruses?

Line 261 It should be clearly stated earlier that cargo proteins are viral gene products. As "cargo" they could be from other sources.

Author Rebuttals to Initial Comments:

We would like to thank the referees for critically reading the manuscript and for making useful suggestions. Point-by-point responses to comments are listed below, with referee comments in italics and amendments and additions to the text highlighted in red. In addition, we have carefully read the manuscript and made several minor changes to improve clarity.

Referee 1

The manuscript by Bayfield et. al., present a cryo-electron microscopy analysis and structural reconstruction of the bacteriophage crAssphage, which is the most abundant virus within the human gut. The importance of crAssphage has recently been highlighted through its dominance of the gut, accounting for upwards of 90% of the virions in certain individuals. However, the biological relevance, function and inner workings of this virus remain largely unresolved. This manuscript provides a high-resolution analysis of the structural components of the crAssphage virion and blends these with a number of unique observations and biological inferences.

I should note that I am not a structural biologist and did not critically assess these aspects of the manuscript, but rather assessed the phage biology and broader relevance of the work. I have no doubts this work will be important to the structural biology field. More broadly, I do think this work has significant impact and interest to warrant consideration by Nature. This is due to the uniqueness and dominance of this virus within the human microbiome and this manuscript makes several novel observations and insights into the structures present within this virion and their putative function. I would like to commend the authors on their writing and the incorporation of biological insights into the manuscript, particularly the sections describing the structure and subsequent function of the muzzle protein and their model for protein and DNA injection. These sections were accessible and provided greater biological insight into the function of crAssphage.

Largely this is an extremely well written, prepared, and pitched manuscript and I have no major comments. I do have two small points for the authors to consider broadening the appeal of their paper outside of the structural biology field.

We thank the referee for the very positive assessment.

Lines 37-38 – When introducing crAssphage, I would recommend the authors consider adding a few more sentences to explain some relevant points on the known crAssphage infection mechanism and lifecycles. Describing host range and some base lifestyle and infection dynamics would bring additional relevance to the later sections.

To address this point, we have added a further description as suggested: “Some crassviruses appear to establish a distinct form of carrier state infection, with delayed lysis of the infected bacteria, following the piggyback-the-winner virus-host dynamic^{6,13}, although how this lifestyle contributes to their prevalence and abundance, remains unclear.”

Lines 109-119 - Is there any information on this fold or closest structural homologue for these two fiber vertex proteins? This is usual for a phage to alternate these capsid display proteins. There was one brief point on potential role of these structures but compared with other sections of the manuscript this section left underdone. Including some additional analyses and biological insight into the potential function of these proteins would further improve.

As requested, we’ve added further analyses in the main text as well as multiple protein alignments to supplementary data, to elaborate on detectable homologues of the fiber proteins. The added text reads: “Homologues of the C-terminal domain of gp21 (residues 42–97) with high sequence similarity were detected in viruses outside of the *Crassvirales* (**Supplementary Data**) including gp8.5 of *Bacillus* phage ϕ 29 (40% identity with residues 226–280 and DALI Z-score of 4.3 and RMSD 3.2 Å between PDB 3QC7 chain A and gp21 AlphaFold structure prediction), **Supplementary Figure 2**. Gp8.5 has been suggested to bind to cell wall factors of gram-positive bacteria¹⁶. AlphaFold modelling showed the C-terminal domain of gp29 (residues 58–118) forms a β -sandwich (**Supplementary Figure 2**) with structural similarity to the Ig-like fold (DALI Z-score of 6.8 and RMSD of 2.1 Å with PDB 2NSM chain A; CATH superfamily 2.60.40.1120). A DALI search for the N-terminal domains of gp21 and gp29 failed to detect any significantly similar structures.”

Referee 2

Overview

In the article, Bayfield et al. “Structural atlas of the most abundant human gut virus,” the authors share the molecular reconstruction of the virion of a tailed bacteriophage representative in the order Crassvirales, a frequent but still relatively unknown group of viruses found in the human gut. The cryo-electron microscopy maps led to a 3D reconstruction of the virion near 3-angstrom resolution facilitating the molecular modeling of all the structural components of the capsid and the tail. The maps also captured the internal organization of the DNA genome and the presence of cargo proteins, which the authors corroborated using mass spectroscopy. The quality of the virion reconstruction is excellent, and the results are significant for the scientific community, as highlighted in my next paragraph. Nonetheless, I found that the interpretation of the results is strongly speculative and overlooks prior results, as pointed out in my major comments below.

Relevance

The study of the crAss001 presents the first high-resolution molecular reconstruction of a representative in the crAss-like phages group, representing an essential scientific contribution to the community. The structural reconstructions will provide valuable insight to infer the structure of the

order of Crassvirales, guiding further research of these frequent but elusive human gut viruses. The molecular information of the tail proteins could guide the inference of host-phage interactions in other viruses of this critical group. Additionally, several findings might particularly interest the structural virology community. The molecular structure of the virion departs significantly from other known tailed phages, for example, the type of auxiliary protein present on the capsid, the size of the tail for a podovirus-like phage, or the intriguing presence (and ejection mechanism) of the cargo proteins. Overall, this is an excellent contribution despite the criticisms added below.

We thank the reviewer for their positive overall assessment.

Major comments

1. The authors claim that the structure suggests "a general mechanism for protein ejection, which involves partial unfolding of proteins during their extrusion through the tail." This interpretation overlooks that the virions were extracted and dialyzed at 4°C (lines 394-404), but the virion infects the host at 37°C (lines 392-393). Previous studies (not cited in the article) have shown that the packaged DNA in other tailed phage model systems, like lambda phage infecting E. coli, undergoes a solid-to-fluid-like transition that facilitates infection near 37°C (Liu et al. PNAS 2014; Evilevitch, eLife 2018). Therefore, one should be cautious in making interpretations of the virion obtained at 4°C. The article should state clearly (and discuss carefully) that the virion corresponds to the structure at 4°C and that, based on prior studies, the internal state of the virion (including the cargo proteins) is likely to change near 37°C. The article should tone down, reduce the length, and remove the figure regarding the current speculation.

As referee #2 suggested, we've significantly shortened the last section describing the model of protein ejection. Although referee #2 also suggested removing the figure depicting a model for ejection, Referee #1 clearly supported its inclusion: *"I would like to commend the authors on their writing and the incorporation of biological insights into the manuscript, particularly ... their model for protein and DNA injection."*

As a compromise, we've moved Fig.6E into Supplementary Information. We've also emphasised that the model described in this section is a hypothesis to be tested: **"understanding how exactly proteins and DNA are ejected requires further studies"**. The discussion of protein ejection in this significantly shortened section is based on previous experimental data on bacteriophages P22 and T7 (references 33, 34, 36) and on the accurate experimental observation of different segments of the same protein, gp45, accommodated in the head and tail, presented for the first time in this study. Whilst we've shortened this section, as requested, we note that the model developed in newly added references 39 and 40 describes a solid-to-fluid-like transition in DNA, and not in protein. For clarity, we amended the title of this section to emphasise our focus on protein ejection. Numerous studies also indicate that protein conformation is largely unaffected by lowering the temperature, and indeed, most structures in the Protein Data Bank (and associated biological models) were derived from proteins purified/stored at 4°C, and vitrified at liquid nitrogen temperatures.

We also note that although the DNA in the core of the capsid may indeed become more conformationally variable at elevated temperatures, as in the model described in Liu et al. PNAS 2014 and Evilevitch eLife 2018, this is unlikely to impact the protein cargo zones that are localised closer to the capsid wall. Furthermore, phage λ is not expected to have the same protein cargoes as crassviruses, and densities inside the capsids of tailed viruses are known to vary widely. Being a siphovirus, λ also has a tail architecture distinct from those of crassviruses.

We are, however, thankful to Referee 2 for raising this thought-provoking point, and note that our model, focusing on protein ejection, does not contradict the possibility of DNA ejection from crassviruses being dependent on temperature. To further clarify this, we made the following addition to the text: “DNA ejection could be facilitated by a solid-to-fluid-like transitions as suggested for bacteriophage λ ^{39,40}.”

2. The article proposes that the muzzle at the end of the tail adopts a new fold, which the authors coined as crAss fold. However, no data regarding the lack of similarity or homology with other proteins are shared (lines 248-249). What were the tests done to arrive at this conclusion? Are there no similarities between the domains of the muzzle and other proteins? What are the closest protein structures and protein topologies known? By how much it differs? The protein fold's quantitative and qualitative assessment should be clear if it is proposed as a new fold. Additionally, a precise figure about these findings is far more necessary in the article than the current speculative ejection mechanism figure.

We thank the referee for this valuable point. Our searches using BLAST, HHPRED and DALI for the muzzle protein gp44 residue segment 453–1012 (560 residues), which contains the crAss domain, returned no significant hits. However, a small 70 amino acid subdomain of gp44 (comprising residues 453–462, 563–571, 577–580, 581–591, 709–716, 725–733, 833–839, 894–901, 902–905) forms a 6-stranded β -barrel that has structural resemblance with 6-stranded β -barrels of other proteins (the lowest RMSD of 3.2 Å is observed for the β -barrel of the ribosomal protein L2, PDB code 4U67). To address this point, we have made the following addition to the text: “A small β -barrel below domain IG2 (indicated by a dotted line in Fig.5C; 70 residues spanning segments 453–462, 563–571, 577–580, 581–591, 709–716, 725–733, 833–839, 894–901, 902–905) structurally resembles other 6-stranded β -barrels (for example, RMSD of 3.2 Å with 50S ribosomal protein L2, PDB code 4U67, chain B).”

3. The title states that the structure reconstructed corresponds to "the most abundant human gut virus." However, it is not evident what they mean by the most abundant virus (crAss001? the whole crAss-like group?). To my understanding, the specific virus investigated, crAss001, is not the most

abundant virus in the human gut. The original crAssphage obtained from uncultured genomes in Dutilh et al. Nature Comm. 2014 (ref. 2) was several times more abundant than all other known phages then. But crAss001 shares little sequence similarity with the original crAssphage. crAss001 belongs to the "group of loosely related bacteriophages termed crAss-like bacteriophages (Yutin et al. Nat. Commun. 2018) (quoted from ref. 6). But what is the actual abundance of crAss001? Additionally, the article does not include a relevant study on human gut viruses: Benler et al. Microbiome 2021. That study investigated human gut metagenomes, identifying new abundant virus groups, such as the Flandesviridae and Quimbyviridae candidate families. These families reached similar detection frequencies as crAss-like phages despite having fewer phages in the database and displaying shorter genomes. Thus, it is not even clear that the crAss-like group is that abundant compared to other new virus groups, let alone crAss001. It is undeniable that the crAss-like phage group is a frequent, cosmopolitan, and intriguing group of human gut viruses. However, any claims regarding their abundance should be more accurate, precise, and cautious in the article.

We thank the referee for raising this question about the abundance of crAssviruses. This comment is partly addressed by Referee 3: "*Bayfield et al present a structural tour de force of the Bacteroides intestinalis bacteriophage Φ crAss001, a prototype of the crAssviruses that are purported to play major roles in shaping the composition and functionality of the human microbiome. Φ crAss001 is the first crAssvirus to be produced in culture, allowing the structural characterization presented here.*"

We now refer to the Benler et al. Microbiome 2021 study, as well as Camarillo-Guererro et al. Cell 2021, and Guerin et al. 2018 regarding variation in relative abundances across cohorts, making the following addition to the text: "*These viruses, (hereafter referred to as crAssviruses), have been closely associated with human populations throughout evolution¹¹. CrAssviruses seem to infect exclusively diverse members of the bacterial phylum Bacteroidota^{3,8,11,12}. In healthy adult Western cohorts, crAssviruses are detected in 98–100% of individuals, often dominating the faecal virome³. The phage families Flandesviridae, Quimbyviridae, and Gubaphage have been identified as close contenders with the CrAssvirales in terms of their detection frequency^{8,12}. However, CrAssvirales appear to be unique in showing both a high abundance in individual metaviromes and across varied cohorts globally^{3,8,11,12}.*"

The abundance of a particular crAssvirus species (in a particular cohort) can only be understood against the backdrop of high abundance that crAssviruses seem able to manifest. It is therefore their common features we look to first. Whilst crAssvirus genomes may indeed present them as a "group of loosely related bacteriophages" (Yutin et al. 2018), their relatedness is well-established and they are now formally recognized to comprise the distinct order CrAssvirales. Here, we show that they possess a contingent of conserved structural features that in all likelihood reflect conserved assembly and infection mechanisms. Φ crAss001 shares the same structural hallmarks with all other crAssvirus groups, as shown in Figure 1D, and the structural atlas is thus applicable to all crAssviruses. To avoid potential ambiguity, we use "viruses" and not "virus" in the revised title.

Referee #3 (all referee comments in italics):

Bayfield et al present a structural tour de force of the Bacteroides intestinalis bacteriophage Φ CrAss001, a prototype of the crassviruses that are purported to play major roles in shaping the composition and functionality of the human microbiome. Φ CrAss001 is the first crassvirus to be produced in culture, allowing the structural characterization presented here.

The authors provide a remarkably detailed picture of the phage structure with the focus on the novelty of some features when compared with other phage structures. The differences include the, so called, muzzle protein rings formed at the distal end of the tail and potentially participating in the phage interaction with the host cell plasma membrane. These gene products appear to be unique to the crassviruses. In addition, there are features of the cargo proteins that appear unusual. The auxiliary capsid proteins are novel in their tertiary structure and possibly their mode of interacting with the major capsid proteins. I am not familiar with other examples of an auxiliary protein covering nearly the entire surface of a particle. A side view in figure 2 D or E would be helpful.

We thank the referee for a supportive overall assessment.

To address the point regarding the novel auxiliary protein, we have added additional panels to Figure 2 showing side views, as suggested. We have also added a supplementary figure to further illustrate the composition and layered structure of the capsid wall. (**Supplementary Figure 1**).

The paper is high density as necessitated by the large number of structural proteins identified and their interactions with neighbors. As the paper flows it is difficult to tell which of the proteins are possibly unique to this virus and which have homologs in other bacteriophages. An example is the portal protein subunit when it is introduced on line 120. It is presented without specifically saying that the structure is homologous to those previously observed and only at line 135 where they begin describing the stem domain and other named features of portal proteins. It would be helpful if, as each protein structure is presented, it is immediately stated that the overall fold is homologous to previously reported proteins. Better yet, in line 57 it could be stated that all of the structures reported except the muzzle proteins and the auxiliary capsid protein have structural homologs in other phages and that this paper reveals the novel insertions and deletions that are the basis for the crassvirus sequence analysis presented in figure 1 C and D.

We thank the referee for these useful suggestions. We have made additions in the text, as suggested. This comment is also addressed in responses to further points made by this referee relating to homology revealed by structural comparison.

It is difficult to fully assess the technical quality of the reconstructions from the material provided for reviewed. The authors describe some remarkably detailed features for the cargo proteins that are probably present at low occupancy. They have undoubtedly applied all of the various tools for

improving local resolution in the presence of high symmetry and this has been effective in discerning subtle differences in quasi-equivalent interactions.

We thank the referee for this comment regarding assessments of data quality. To address this we have expanded Table S1 containing data collection and refinement statistics. Further details, including FSC curves, are listed in validation reports created during data deposition with the wwPDB. These files have been uploaded along with the revised manuscript, and we requested release of all structural data along with the reports via PDB immediately following publication. To illustrate data quality, we've added **Supplementary Figure 6**, showing regions of protein structures with corresponding density maps for all proteins including the cargo protein gp45. Maps and models were also uploaded along with the revised version.

Below are a few points that the authors should address.

Line 82 There should be a reference to the HK97 capsid protein fold.

A reference has now been added.

Line 84 Is the E-loop insertion novel or are there comparable insertions in non crass phage?

This has been addressed by the following addition to the text: *"An insertion at a similar position of the E-loop, albeit of a different fold, is present in the major capsid protein of bacteriophage T4¹⁵. The α crAss001 I-domain is structurally similar to the I-domain of the major capsid protein of phage P22 (DALI Z-score of 6.1 and RMSD 2.7 Å with PDB entry 5UU5 chain E). However, in P22, this domain (residues 226–344) is part of the A-domain, rather than the E-loop."*

Line 86 Gp36 appears to have a different association with the capsid when compared to other auxiliary proteins. As mentioned in the text it is unusual for an auxiliary protein to cover so much of the particles. It appears to interact with the MCP in multiple places. A side view in Fig. 2 C or D would clarify this.

Figure 2 has been expanded in response to Referee's earlier comment.

Line 112 Are structural homologs of the fiber proteins present in other phages?

The same point has been raised by Referee 1; we have addressed this above.

Line 128 Should say whether the portal protein has the canonical fold and a reference to the portal protein structure should be included.

To address this, we made the following addition to the text: “The portal protein exhibits the canonical fold (Fig. 3A)²¹, with the long C-terminal barrel of the oligomer resembling the corresponding domain of the P22 portal protein²².”

Line 138 Similar structure in P22 Tang J, etal Structure. 2011 Apr 13;19(4):496-502.

This has now been addressed in replying to the comment above.

Line 144 Does this mean that no symmetry was applied. State this clearly.

Symmetry was not applied and this has now been clarified in the text.

Line 190 What is the sequence identity among Ring proteins 1.2.3 and 4/5?

This has been now stated in the text: “Despite their conserved fold, the sequence identity among the ring proteins is relatively low, ranging from 14.9% (R2-R3) to 18.6% (R1-R2), although the more closely related paralogues in Φ crAss001, R3 and R4, are 30.6% identical.”

Line 196 All rings are formed by the same gene product?

This has now been clarified in the text.

Line 200 varying number of rings in other viruses?

This has been addressed in the text.

Line 261 It should be clearly stated earlier that cargo proteins are viral gene products. As “cargo” they could be from other sources.

We have clarified this in the text.

Reviewer Reports on the First Revision:

Referee #1 (Remarks to the Author):

I'd like to thank the authors for their revised manuscript and for addressing my previous comments. I have no further requested changes.

Referee #2 (Remarks to the Author):

The authors addressed the comments in my initial report. Below I added two suggestions about the revised version of the manuscript.

In Line 322, "Assuming proteins were packed at a density $2.15 \text{ \AA}^3/\text{Da}$," replace "density" with "volume per molecular weight" or give density in units of $\text{Da}/\text{\AA}^3$.

Regarding the new title, the use of "viruses" suggests that the publication includes a structural atlas of more than one virus, which is not the case. Therefore, I recommend the authors revise the title to be accurate, for example, "Structural atlas of a representative from one of the most abundant human gut viral groups."

Referee #3 (Remarks to the Author):

The revised version of the manuscript has, for the most part, incorporated my suggestions and addressed my concerns. The only point of clarification would be to add

Line 22 Virally encoded cargo proteins

Line 52 virally encoded

I realize that I appear fixated on this point, but there are now numerous examples of viruses picking up host gene products and my first reading of the manuscript made that wrong assumption.

This is a remarkable contribution to the phage literature and I applaud the authors for their work.

Author Rebuttals to First Revision:

Responses to Referees' comments:

Referee 1

I'd like to thank the authors for their revised manuscript and for addressing my previous comments. I have no further requested changes.

We are grateful for the reviewer's comments, which helped to improve the manuscript.

Referee 2

The authors addressed the comments in my initial report. Below I added two suggestions about the revised version of the manuscript.

In Line 322, "Assuming proteins were packed at a density 2.15 A³/Da," replace "density" with "volume per molecular weight" or give density in units of Da/A³.

The text has been changed to "volume per molecular weight".

Regarding the new title, the use of "viruses" suggests that the publication includes a structural atlas of more than one virus, which is not the case. Therefore, I recommend the authors revise the title to be accurate, for example, "Structural atlas of a representative from one of the most abundant human gut viral groups."

While the "atlas" includes comparative genomic analysis, and thus spans multiple crassvirus groups, we have chosen to use a version of the Editor's suggested title in addressing this comment.

Referee 3

The revised version of the manuscript has, for the most part, incorporated my suggestions and addressed my concerns. The only point of clarification would be to add

Line 22 Virally encoded cargo proteins

Line 52 virally encoded

I realize that I appear fixated on this point, but there are now numerous examples of viruses picking up host gene products and my first reading of the manuscript made that wrong assumption.

We have amended “cargo proteins” to “virally encoded cargo proteins”.

This is a remarkable contribution to the phage literature and I applaud the authors for their work.

We are grateful for the reviewer’s positive assessment and their constructive critique throughout.